# δ Gravity: Dark Sector, Post-Newtonian Limit and Schwarzschild Solution

**Jorge Alfaro** [1,*,†] and **Pablo González** [2,†]

1 Facultad de Física, Pontificia Universidad Católica de Chile, Casilla 306, Macul, Santiago 7810000, Chile
2 Sede Esmeralda, Universidad de Tarapacá, Avda. Luis Emilio Recabarren 2477, Iquique 1130000, Chile; pgonzalezv@uta.cl
\* Correspondence: jalfaro@fis.puc.cl
† These authors contributed equally to this work.

**Abstract:** We present a new kind of model, which we call δ Theories, where standard theories are modified including new fields, motivated by an additional symmetry (δ symmetry). In previous works, we proved that δ Theories just live at one loop, so the model in a quantum level can be interesting. In the gravitational case, we have δ Gravity, based on two symmetric tensors, $g_{\mu\nu}$ and $\tilde{g}_{\mu\nu}$, where quantum corrections can be controlled. In this paper, a review of the classical limit of δ Gravity in a Cosmological level will be developed, where we explain the accelerated expansion of the universe without Dark Energy and the rotation velocity of galaxies by the Dark Matter effect. Additionally, we will introduce other phenomenon with δ Gravity like the deflection of the light produced by the sun, the perihelion precession, Black Holes and the Cosmological Inflation.

**Keywords:** modified gravity; dark energy; dark matter; general relativity

## 1. Introduction

In the last century, the cosmological observations have revealed that the dynamic of the Universe is dominated by an accelerated expansion, apparently due to a mysterious component called Dark Energy [1–3]. Additionally, these observations say that most of the matter is in the form of unknown *particles* that interact principally by gravitation with the ordinary matter, called Dark Matter [4]. In this paper, we will refer to Dark Matter (DM) and Dark Energy (DE) effects as the Dark Sector. Although General Relativity (GR) is able to accommodate the Dark Sector, its interpretation in terms of fundamental theories of elementary particles is problematic [5].

For one side, some candidates exist that could play the role of DM, but none have been detected yet. In a galactic scale, DM produces an anomalous rotation velocity which is relatively constant far from the center of the galaxy [6–12], and a lot of alternative models, where a modification to gravity is introduced, have been developed to explain this effect. For instance, an explanation based on the modification of the dynamics for small accelerations cannot be ruled out [13,14].

On the other side, DE can be explained with a small cosmological constant (Λ). At early times, Λ is not important for the evolution of the Universe, but at later stages it will dominate the expansion, explaining the observed acceleration. However, the cosmological constant is too small to be generated in Quantum Field Theory models because it is the vacuum energy, which is usually predicted to be very large [15].

For all these reasons, one of the most important goals in cosmology and cosmic structure formation is to understand the Dark Sector in the context of a fundamental physical theory [16,17]. Some explanations include additional fields in approaches like quintessence, chameleon, vector Dark Energy or massive gravity; the addition of higher order terms in the Einstein–Hilbert action, like $f(R)$

theories and Gauss–Bonnet terms and finally the introduction of extra dimensions for a modification of gravity on large scales (see, for instance, [18]).

Recently, in [19], a model of gravitation that is very similar to GR is presented. In that paper, two different points were considered. The first is that GR is finite on shell at one loop in vacuum [20], so renormalization is not necessary at this level. The second is the approach of $\delta$ gauge theories (DGT), studied for the first time in [21,22] and defined as follows: (a) consider a model described by a set of fields $\phi_I$ and an action $S(\phi_I)$, which is invariant under an algebra of infinitesimal transformations $G$. (b) A new kind of field $\tilde{\phi}_I$ is introduced, different from the original set $\phi_I$. In the extended set of fields, an extra symmetry that we call $\delta$ symmetry is realized. It is formally obtained as the variation of the original symmetry $G$. (c) We find an action for the extended set of fields which is invariant under the extended symmetry. The action is unique if we impose the additional condition that the original action for $\phi_I$ is recovered if all $\tilde{\phi}_I$ vanish. (c) It turns out that the classical equations of motion of $\phi_I$ with the action $S(\phi_I)$ are still satisfied, even in the quantum level where the corrections live only at one loop. Therefore, when we implement this extension to General Coordinates Transformations (GCT), which we call Extended General Coordinates Transformations (ExGCT), and imposing the extended symmetry provides a unique extension for the Einstein–Hilbert action, which define the dynamics of the new model, called in this paper $\delta$ Gravity. For these reasons, the original motivation was to develop the quantum properties of this model (see [19]). In this work, we will study the classical properties of $\delta$ Gravity.

A first approximation was developed in Section [23,24], presenting a truncated version of $\delta$ Gravity applied to Cosmology. The $\delta$ symmetry was fixed in different ways in order to simplify the analysis of the model and explain the accelerated expansion of the universe without DE. The results were quite reasonable taking into account the simplifications involved, but $\delta$ Matter was ignored in the process. After in [25], we developed the Cosmological solution in a $\delta$ Gravity version where $\delta$ symmetry is preserved, which means that we are forced to include $\delta$ Matter. In that case, the accelerated expansion can be explained in the same way and additionally we have a new component of matter as DM candidate. In addition, we guaranteed that the special properties of $\delta$ Theories previously mentioned are preserved. In this work, we will continue with the full-fledged $\delta$ Gravity presented in [25].

We will develop this paper as follows. In Section 2, a complete resume about the bases of $\delta$ Theories will be presented. Then, we will show the $\delta$ Gravity action that is invariant under ExGCT, including the equations of motion. We will see that the Einstein's equations with the usual Energy momentum Tensor, $T_{\mu\nu}$, continue to be valid, but new equations of motion appears for $\tilde{g}_{\mu\nu}$, depending on a new Energy momentum Tensor, $\tilde{T}_{\mu\nu}$, by the presence of $\delta$ Matter. Additionally, we will derive the equation of motion for the test particle. We distinguish the massive case, where the equation is not a geodesic, and the massless case, where we have a null geodesic with an effective metric. A complete derivation of all these is presented in [23,24].

To give a complete description of $\delta$ Gravity in this paper, in Section 3, we will present the cosmological case developed in [25]. We obtain the accelerated expansion of the universe assuming a universe without DE, i.e., only having non-relativistic matter and radiation which satisfy a fluid-like equation $p = \omega\rho$, for both normal and $\delta$ Matter. A preliminary computation was done in [23], where an approximation is discussed. Later, in [24], we developed an exact solution of the equations, but in both cases we assumed that we do not have $\delta$ Matter, such that the new symmetry is broken. For that reason, in [25], we studied $\delta$ Gravity with $\delta$ Matter to preserve the new symmetry. This is very interesting because the presence of $\delta$ Matter gives us a possibility to explain DM. In both limits, the solution is used to fit the supernovae data and get predictions for the cosmological parameters. In this calculation, we obtained that the physical reason for the accelerated expansion of the universe is a geometric effect, where an effective scale factor is defined by the model, producing the accelerated expansion of the universe without a Cosmological Constant and predicting a Big-Rip as [26–30]. This effective scale factor agrees with the standard cosmology at early times and shows acceleration

only at late times. Therefore, we expect that primordial density perturbations should not have large corrections. Moreover, in [25], the value of the cosmological parameters improve greatly by the inclusion of $\delta$ Matter, increasing the importance of this component. For instance, the age of the Universe is much closer to the Planck satellite value now than the value we got in [24]. Finally, we will include a brief analysis of the Inflation Case to present how $\delta$ Gravity could explain the exponential expansion in this era just like the accelerated expansion by DE.

To study the DM phenomenon and complete the Dark Sector in $\delta$ Gravity, in Section 4, we will study the Non-Relativistic case to understand the behavior of the model in the (Post)-Newtonian limit. We have that $\delta$ Gravity agrees with GR at the classical level far from the sources. However, inside a matter fluid like a galaxy, new effects appear because of $\delta$ Matter. In that sense, $\delta$ Matter could be considered like a DM candidate. In this limit, a relation between the ordinary density and $\delta$ Matter density will be found. With this and some realistic density profiles as Einasto and Navarro–Frenk–White (NFW) profiles for a galaxy [31–36], we can study the modifications in the rotation velocity. We will see that $\delta$ Matter effect is not related to the scale, but rather to the behavior of the distribution of ordinary matter. In the solar system scale, where the large structures as planets and stars have a concentrated and almost constant distributions, $\delta$ Matter is practically negligible. However, in a galactic scale, where the distribution is strongly dynamic, $\delta$ Matter will be important to explain the DM effect. Thus, the amount of ordinary DM could be much less, explaining its extremely problematic detection.

We know that the causal structure of $\delta$ Gravity in vacuum is the same as in GR. However, important effects could appear when really massive object, as Black Holes, are taken into account. Thus, in Section 5, we will solve the equation of motion of $g_{\mu\nu}$ and $\tilde{g}_{\mu\nu}$ for the Schwarzschild Case in the vacuum with appropriate boundary conditions. Then, we will use this solution to compute the deflection of light by the sun and analyze the perihelion precession [37]. We have to guarantee that these results are very close to GR, unless we consider highly massive object.

We have to say that models like $\delta$ Gravity are not ghost-free, which can produce some problems like non-unitarity or instabilities [22,38–41]. This ghost component produces a phantom behaviour. A scalar phantom has been used in the current literature to describe the current data of the most classical tests of cosmology [26–30], but this background solution becomes unstable by the ghost field. Without a doubt, the ghost problem must be kept in mind, but some solutions can be implemented to obtain harmless ghosts (see, for instance, [42–46]). In spite of the ghost problem, the nature of the Dark Sector is such an important and difficult problem that cosmologists do not expect to solve in one stroke, so we must be open to explore new possibilities. For that reason, we will study $\delta$ Gravity as a classical effective model and use it in Cosmology. This means to approach the problem from the phenomenological side instead of neglecting it a priori because it does not satisfy yet all the properties of a fundamental quantum theory. Now, the phantom problem is being studied in this moment for $\delta$ Gravity and the results will be presented in a future work.

With respect to the extended symmetry, we must emphasize that a closed algebra is formed (see [47]), producing a Noether conserved current to define the usual Energy-Momentum Tensor $T_{\mu\nu}$ with the usual conservation equation and another one with a new Energy-Momentum Tensor $\tilde{T}_{\mu\nu}$, representing the $\delta$ Matter component. In addition, to solve the equations of $g_{\mu\nu}$ and $\tilde{g}_{\mu\nu}$, we need to fix an extended harmonic gauge due to the existence of the GCT and ExGCT symmetries. The closure of the algebra, the ensuing Noether currents and the gauge fixing show that ExGCT is an extension of GCT. The existence of this extra symmetry is crucial to the model. Finally, it should be remarked that $\delta$ Gravity is not a metric model of gravity because only massless particles move on null geodesics, given by a linear combination of both tensor fields. For this, see Section 2.3.

## 2. $\delta$ Gravity

In this paper, we are studying $\delta$ Gravity at a classical level. This gravity model was developed from a special kind of modified theories named $\delta$ Theories. In this section, we will define the $\delta$ Theories in general and their properties and then we will apply this modification to the Einstein–Hilbert model. For more details, see [19,24,47].

### 2.1. $\delta$ Theories Formalism and Modified Action

These modified theories consist of the application of a variation represented by $\tilde{\delta}$. As a variation, it will have all the properties of the usual variation such as:

$$
\begin{aligned}
\tilde{\delta}(AB) &= \tilde{\delta}(A)B + A\tilde{\delta}(B), \\
\tilde{\delta}\delta A &= \delta\tilde{\delta}A, \\
\tilde{\delta}(\Phi_{,\mu}) &= (\tilde{\delta}\Phi)_{,\mu},
\end{aligned}
\tag{1}
$$

where $\delta$ is another variation. The particular point with this variation is, when we apply it on a field (function, tensor, etc.), it will give new elements that we define as $\delta$ fields, which is an entirely new independent object from the original, $\tilde{\Phi} = \tilde{\delta}(\Phi)$. We use the convention that a tilde tensor is equal to the $\delta$ transformation of the original tensor when all its indexes are covariant, which is:

$$
\tilde{S}_{\mu\nu\alpha\ldots} \equiv \tilde{\delta}\left(S_{\mu\nu\alpha\ldots}\right)
\tag{2}
$$

and we raise and lower indexes using the metric $g_{\mu\nu}$. Therefore:

$$
\begin{aligned}
\tilde{\delta}\left(S^{\mu}{}_{\nu\alpha\ldots}\right) &= \tilde{\delta}(g^{\mu\rho}S_{\rho\nu\alpha\ldots}) \\
&= \tilde{\delta}(g^{\mu\rho})S_{\rho\nu\alpha\ldots} + g^{\mu\rho}\tilde{\delta}\left(S_{\rho\nu\alpha\ldots}\right) \\
&= -\tilde{g}^{\mu\rho}S_{\rho\nu\alpha\ldots} + \tilde{S}^{\mu}{}_{\nu\alpha\ldots},
\end{aligned}
\tag{3}
$$

where we used that $\delta(g^{\mu\nu}) = -\delta(g_{\alpha\beta})g^{\mu\alpha}g^{\nu\beta}$. Now, with the previous notation in mind, we can define how the tilde elements, given by Equation (2), transform. In general, we can represent a transformation of a field $\Phi_i$ like:

$$
\bar{\delta}\Phi_i = \Lambda_i^j(\Phi)\epsilon_j,
\tag{4}
$$

where $\epsilon_j$ is the parameter of the transformation. Then, $\tilde{\Phi}_i = \tilde{\delta}\Phi_i$ transforms:

$$
\bar{\delta}\tilde{\Phi}_i = \tilde{\Lambda}_i^j(\Phi)\epsilon_j + \Lambda_i^j(\Phi)\tilde{\epsilon}_j,
\tag{5}
$$

where we used that $\tilde{\delta}\bar{\delta}\Phi_i = \bar{\delta}\tilde{\delta}\Phi_i = \bar{\delta}\tilde{\Phi}_i$ and $\tilde{\epsilon}_j = \tilde{\delta}\epsilon_j$ is the parameter of the new transformation. For example, if we consider General Coordinates Transformation (GCT) or diffeomorphism in its infinitesimal form, we have:

$$
x'^{\mu} = x^{\mu} - \xi_0^{\mu}(x) \quad \rightarrow \quad \bar{\delta}x^{\mu} = -\xi_0^{\mu}(x)
\tag{6}
$$

and defining:

$$
\xi_1^{\mu}(x) \equiv \tilde{\delta}\xi_0^{\mu}(x),
\tag{7}
$$

we can use Equation (5) to see a few examples of how some elements transform:

**(I)** A scalar $\phi$:

$$\bar{\delta}\phi = \xi_0^\mu \phi_{,\mu}, \tag{8}$$

$$\bar{\delta}\tilde{\phi} = \xi_1^\mu \phi_{,\mu} + \xi_0^\mu \tilde{\phi}_{,\mu}. \tag{9}$$

**(II)** A vector $V_\mu$:

$$\bar{\delta}V_\mu = \xi_0^\beta V_{\mu,\beta} + \xi_{0,\mu}^\alpha V_\alpha, \tag{10}$$

$$\bar{\delta}\tilde{V}_\mu = \xi_1^\beta V_{\mu,\beta} + \xi_{1,\mu}^\alpha V_\alpha + \xi_0^\beta \tilde{V}_{\mu,\beta} + \xi_{0,\mu}^\alpha \tilde{V}_\alpha. \tag{11}$$

**(III)** Rank two Covariant Tensor $M_{\mu\nu}$:

$$\bar{\delta}M_{\mu\nu} = \xi_0^\rho M_{\mu\nu,\rho} + \xi_{0,\nu}^\beta M_{\mu\beta} + \xi_{0,\mu}^\beta M_{\nu\beta}, \tag{12}$$

$$\bar{\delta}\tilde{M}_{\mu\nu} = \xi_1^\rho M_{\mu\nu,\rho} + \xi_{1,\nu}^\beta M_{\mu\beta} + \xi_{1,\mu}^\beta M_{\nu\beta} + \xi_0^\rho \tilde{M}_{\mu\nu,\rho} + \xi_{0,\nu}^\beta \tilde{M}_{\mu\beta} + \xi_{0,\mu}^\beta \tilde{M}_{\nu\beta}. \tag{13}$$

This means that all the fields have $\delta$ partner and their transformations depend on their nature (scalar, vector, tensor, etc.). Particularly, in gravitation, we will have a model with two tensor fields. The first one is just the usual gravitational field $g_{\mu\nu}$ and the second one will be $\tilde{g}_{\mu\nu}$. Then, we will have two gauge transformations associated with GCT. We will call it Extended General Coordinate Transformation (ExGCT), given by:

$$\bar{\delta}g_{\mu\nu} = \xi_{0\mu;\nu} + \xi_{0\nu;\mu}, \tag{14}$$

$$\bar{\delta}\tilde{g}_{\mu\nu} = \xi_{1\mu;\nu} + \xi_{1\nu;\mu} + \tilde{g}_{\mu\rho}\xi_{0,\nu}^\rho + \tilde{g}_{\nu\rho}\xi_{0,\mu}^\rho + \tilde{g}_{\mu\nu,\rho}\xi_0^\rho. \tag{15}$$

With all these, we can introduce the $\delta$ Theories. We start by considering a model that is based on a given action $S_0[\Phi_I]$, where $\Phi_I$ are generic fields. Then, we add to it a piece which is equal to a $\delta$ variation with respect to the fields and we let $\tilde{\delta}\Phi_J = \tilde{\Phi}_J$, so that we have:

$$S[\Phi, \tilde{\Phi}] = S_0[\Phi] + \int d^4x \frac{\delta S_0(\Phi)}{\delta\Phi_I(x)}\tilde{\Phi}_I(x), \tag{16}$$

where the indexes $I$ can represent any kind of indexes. Then, the Action in (16) is invariant under transformations given by (4) and (5), if $S_0[\Phi]$ is invariant under (4). A first important property of this action is that the classical equations of the original fields are preserved. We can see this when Equation (16) is varied with respect to $\tilde{\Phi}_I$:

$$\frac{\delta S_0(\Phi)}{\delta\phi_I} = 0. \tag{17}$$

Obviously, we have new equations when varied with respect to $\Phi_I$. These equations determine $\tilde{\Phi}_I$ and they can be reduced to:

$$\int d^4x \frac{\delta^2 S_0(\Phi)}{\delta\Phi_I(y)\delta\Phi_J(x)}\tilde{\Phi}_J(x) = 0, \tag{18}$$

where $\frac{\delta^2 S_0}{\delta\Phi_I(y)\delta\Phi_J(x)}$ has to be considered as an operation on $\tilde{\Phi}_J$, so the solution of this equation is not trivial. Now, we will apply this result to ExGCT, given by Equations (8)–(15), for gravity.

## 2.2. δ Gravity Action and Equations of Motion

In this paper, we will use the $\delta$ Theories formalism on the Einstein–Hilbert Action to obtain our modified action of gravity, which we call $\delta$ Gravity. That is:

$$S_0 \quad = \quad \int d^4x \sqrt{-g} \left( \frac{R}{2\kappa} + L_M \right), \tag{19}$$

$$S \quad = \quad \int d^4x \sqrt{-g} \left( \frac{R}{2\kappa} + L_M - \frac{1}{2\kappa} \left( G^{\alpha\beta} - \kappa T^{\alpha\beta} \right) \tilde{g}_{\alpha\beta} + \tilde{L}_M \right), \tag{20}$$

where $\kappa = \frac{8\pi G}{c^2}$, $L_M = L_M(\phi_I, \partial_\mu \phi_I)$ is the Lagrangian of the matter fields $\phi_I$, $\tilde{g}_{\mu\nu} = \tilde{\delta} g_{\mu\nu}$, $T^{\mu\nu} = \frac{2}{\sqrt{-g}} \frac{\delta(\sqrt{-g} L_M)}{\delta g_{\mu\nu}}$ and $\tilde{L}_M = \tilde{\phi}_I \left( \frac{\delta L_M}{\delta \phi_I} \right) + (\partial_\mu \tilde{\phi}_I) \left( \frac{\delta L_M}{\delta(\partial_\mu \phi_I)} \right)$, where $\tilde{\phi}_I = \tilde{\delta} \phi_I$ are the $\delta$ Matter fields. Previously, we mentioned that a truncated version of $\delta$ Gravity was presented in [23,24] and applied to Cosmology. To simplify the analysis of the model, we did not consider the $\delta$ Matter components, which means $\tilde{L}_M = 0$, breaking the ExGCT. In this work, we present the full-fledged $\delta$ Gravity, introducing the $\delta$ Matter in order to preserve the $\delta$ symmetry. In this case, the equations of motion are:

$$G^{\mu\nu} \quad = \quad \kappa T^{\mu\nu}, \tag{21}$$

$$F^{(\mu\nu)(\alpha\beta)\rho\lambda} D_\rho D_\lambda \tilde{g}_{\alpha\beta} + \frac{1}{2} g^{\mu\nu} R^{\alpha\beta} \tilde{g}_{\alpha\beta} - \frac{1}{2} \tilde{g}^{\mu\nu} R \quad = \quad \kappa \tilde{T}^{\mu\nu}, \tag{22}$$

with $\tilde{T}_{\mu\nu} = \tilde{\delta} T_{\mu\nu}$ and:

$$F^{(\mu\nu)(\alpha\beta)\rho\lambda} \quad = \quad P^{((\rho\mu)(\alpha\beta))} g^{\nu\lambda} + P^{((\rho\nu)(\alpha\beta))} g^{\mu\lambda} - P^{((\mu\nu)(\alpha\beta))} g^{\rho\lambda} - P^{((\rho\lambda)(\alpha\beta))} g^{\mu\nu},$$

$$P^{((\alpha\beta)(\mu\nu))} \quad = \quad \frac{1}{4} \left( g^{\alpha\mu} g^{\beta\nu} + g^{\alpha\nu} g^{\beta\mu} - g^{\alpha\beta} g^{\mu\nu} \right),$$

where $(\mu\nu)$ denotes that $\mu$ and $\nu$ are in a totally symmetric combination. An important fact to notice is that the Einstein equations, given by Equation (21), are preserved. Thus, the only difference with General Relativity (GR) are the additional equations presented in (22) to find $\tilde{g}_{\mu\nu}$. Moreover, our equations are of second order in derivatives, which is needed to preserve causality. Finally, we have that the action (20) is invariant under the transformations (14) and (15), so two conservation rules are satisfied. They are:

$$D_\nu T^{\mu\nu} \quad = \quad 0, \tag{23}$$

$$D_\nu \tilde{T}^{\mu\nu} \quad = \quad \frac{1}{2} T^{\alpha\beta} D^\mu \tilde{g}_{\alpha\beta} - \frac{1}{2} T^{\mu\beta} D_\beta \tilde{g}^\alpha_\alpha + D_\beta (\tilde{g}^\beta_\alpha T^{\alpha\mu}). \tag{24}$$

They are related to the Noether conserved current of the extended symmetry. Then, we have two Energy-Momentum Tensors, the usual $T_{\mu\nu}$ and $\tilde{T}_{\mu\nu}$. The last one is due to the new symmetry and $\delta$ Matter. Thus, to solve the equations of motion of $\delta$ Gravity, given by (21)–(24), we need an expression of the perfect fluid's energy momentum tensors, given by (A24) and (A25) in Appendix A.

## 2.3. Test Particle

In the previous subsection, we found the equations of motion for $\delta$ Gravity. However, we need to know how the new fields affect the trajectory of a test particle. For this, we will study the test particle action separately for massive and massless particles. The first discussion of this issue in $\delta$ Gravity is in [23].

2.3.1. Massive Particles

In GR, the action for a test particle, including the Einstein–Hilbert term, is given by:

$$S_0[x, g] = \frac{1}{2\kappa} \int d^4 y \sqrt{-g} R - m \int dt \sqrt{-g_{\mu\nu}(x) \dot{x}^\mu \dot{x}^\nu}, \tag{25}$$

with $\dot{x}^\mu = \frac{dx^\mu}{dt}$ and the variation in $x^\mu$ produces the geodesic equation. This action is invariant under reparametrizations, $t' = t - \epsilon(t)$, where the infinitesimal form is:

$$\delta_R x^\mu = \dot{x}^\mu \epsilon, \tag{26}$$

and the action (25) is similar to (19), where $x^\mu$ is a field in $L_M$ in a sense. To obtain the modified action, we have to use Equation (20), where a new field should be obtained, given by $\tilde{\delta} x^\mu$. However, this new field does not make sense because it should represent an additional coordinate system and our model just lives in four dimensions. Therefore, we will impose that $x^\mu$ does not have a $\delta$ partner, so the action in (25) will be modified to:

$$S[X, g, \tilde{g}] = \frac{1}{2\kappa} \int d^4 y \sqrt{-g} R - m \int dt \sqrt{-g_{\mu\nu}(x) \dot{x}^\mu \dot{x}^\nu} - \frac{1}{2\kappa} \int d^4 y \sqrt{-g} \left( G_{\mu\nu} - \kappa T_{\mu\nu}(x) \right) \tilde{g}^{\mu\nu}, \tag{27}$$

where:

$$T_{\mu\nu}(x) = \frac{m}{\sqrt{-g}} \int dt \frac{\dot{x}_\mu \dot{x}_\nu}{\sqrt{-g_{\alpha\beta}(x) \dot{x}^\alpha \dot{x}^\beta}} \delta(y - x). \tag{28}$$

Notice that $T_{\mu\nu}$ is $t$-parametrization invariant, but our action breaks the ExGCT symmetry. Now, extracting the pure $\delta$ Gravity components from the action (27), we obtain the modified test particle action:

$$S_p = m \int dt \left( \frac{\left( g_{\mu\nu} + \frac{1}{2} \tilde{g}_{\mu\nu} \right) \dot{x}^\mu \dot{x}^\nu}{\sqrt{-g_{\alpha\beta} \dot{x}^\alpha \dot{x}^\beta}} \right). \tag{29}$$

This action for a test particle in a gravitational field is the starting point for the physical interpretation of this model. It is t-reparametrization invariant, invariant under general coordinate transformations, but it is not invariant under ExGCT.

Now, the trajectory of massive test particles is given by the equation of motion of $x^\mu$. This equation say us that $g_{\mu\nu} \dot{x}^\mu \dot{x}^\nu = cte$, just like GR. Now, if we choose $t$ equal to the proper time, then $g_{\mu\nu} \dot{x}^\mu \dot{x}^\nu = -1$ and the equation of motion is reduced in this case to:

$$\hat{g}_{\mu\nu} \ddot{x}^\nu + \hat{\Gamma}_{\mu\alpha\beta} \dot{x}^\alpha \dot{x}^\beta = \frac{1}{4} \tilde{K}_{,\mu}, \tag{30}$$

with:

$$\hat{\Gamma}_{\mu\alpha\beta} = \frac{1}{2} \left( \hat{g}_{\mu\alpha,\beta} + \hat{g}_{\beta\mu,\alpha} - \hat{g}_{\alpha\beta,\mu} \right),$$

$$\hat{g}_{\alpha\beta} = \left( 1 + \frac{1}{2} \tilde{K} \right) g_{\alpha\beta} + \tilde{g}_{\alpha\beta},$$

$$\tilde{K} = \tilde{g}_{\alpha\beta} \dot{x}^\alpha \dot{x}^\beta.$$

Equation (30) is a second order equation, but it is not a classical geodesic, because we have additional terms and an effective metric can not be defined. Moreover, the equation of motion is independent of the mass of the particle, so all particles will fall with the same acceleration.

### 2.3.2. Massless Particles

The massless case is particularly important in this work because we need to study photons trajectories to define distances. Unfortunately, the action (25) is useless for massless particles because it is null when $m = 0$. To solve this problem, it is a common practice to start from the action [48]:

$$S_0[\dot{x}, g, v] = \frac{1}{2} \int dt \left( vm^2 - v^{-1} g_{\mu\nu} \dot{x}^\mu \dot{x}^\nu \right), \tag{31}$$

where $v$ is an auxiliary field. From Equation (31), we can obtain the equation of motion for $v$:

$$v = -\frac{\sqrt{-g_{\mu\nu} \dot{x}^\mu \dot{x}^\nu}}{m}. \tag{32}$$

If we substitute Equation (32) in (31), we recover the action (25). This means that (31) is equivalent to (25), but additionally includes the massless case.

In our case, a suitable action, similar to (31), is:

$$S[\dot{x}, g, \tilde{g}, v] = \int dt \left( m^2 v - \frac{(g_{\mu\nu} + \tilde{g}_{\mu\nu}) \dot{x}^\mu \dot{x}^\nu}{4v} + \frac{m^2 v^3}{4 g_{\alpha\beta} \dot{x}^\alpha \dot{x}^\beta} \left( m^2 + v^{-2} \tilde{g}_{\mu\nu} \dot{x}^\mu \dot{x}^\nu \right) \right). \tag{33}$$

In $\delta$ Gravity, the equation of $v$ is still (32), and if we use it in (33), we obtain the massive test particle action given by (29). However, now, we can study the massless case.

If we evaluate $m = 0$ in (31) and (33), we can compare GR and $\delta$ Gravity, respectively. They are:

$$S_0^{(m=0)}[\dot{x}, g, v] = -\frac{1}{2} \int dt v^{-1} g_{\mu\nu} \dot{x}^\mu \dot{x}^\nu, \tag{34}$$

$$S^{(m=0)}[\dot{x}, g, \tilde{g}, v] = -\frac{1}{4} \int dt v^{-1} \mathbf{g}_{\mu\nu} \dot{x}^\mu \dot{x}^\nu, \tag{35}$$

with $\mathbf{g}_{\mu\nu} = g_{\mu\nu} + \tilde{g}_{\mu\nu}$. In both cases, the equation of motion for $v$ implies that a massless particle move in a null-geodesic. In the usual case, we have $g_{\mu\nu} \dot{x}^\mu \dot{x}^\nu = 0$. However, in our model, the null-geodesic is given by $\mathbf{g}_{\mu\nu} \dot{x}^\mu \dot{x}^\nu = 0$, so the trajectory obey an effective metric given by $\mathbf{g}_{\mu\nu} = g_{\mu\nu} + \tilde{g}_{\mu\nu}$. The equation of motion for the path of a test massless particle is given by:

$$\mathbf{g}_{\mu\nu} \ddot{x}^\nu + \mathbf{\Gamma}_{\mu\alpha\beta} \dot{x}^\alpha \dot{x}^\beta = 0, \tag{36}$$
$$\mathbf{g}_{\mu\nu} \dot{x}^\mu \dot{x}^\nu = 0,$$

with:

$$\mathbf{\Gamma}_{\mu\alpha\beta} = \frac{1}{2} (\mathbf{g}_{\mu\alpha,\beta} + \mathbf{g}_{\beta\mu,\alpha} - \mathbf{g}_{\alpha\beta,\mu}).$$

On the other side, in [19], a quantum analysis of $\delta$ Gravity was presented, where the quantum corrections just live at one loop only, producing an interesting chance to obtain a quantum gravity model. In that work, we did a counting of degrees of freedom, where the combination $g_{\mu\nu} + \tilde{g}_{\mu\nu}$ is a normal particle, whereas $\tilde{g}_{\mu\nu}$ is a ghost. This result and the fact that the effective metric $\mathbf{g}_{\mu\nu} = g_{\mu\nu} + \tilde{g}_{\mu\nu}$ defines the geometry in our model say that this particular combination must be seen as the unique graviton [38–41].

Additionally, we know that the proper time must be defined for massive particles. The equation of motion for massive particles satisfies the important property of preserving the form of the proper time in a particle in free fall. Notice that, in our case, the quantity that is constant using the equation of

motion for massive particles, derived from Equation (30), is $g_{\mu\nu}\dot{x}^{\mu}\dot{x}^{\nu}$. This single out this definition of proper time and not other. Thus, we must define proper time using the original metric $g_{\mu\nu}$. That is:

$$g_{\mu\nu}\left(\frac{1}{c}\frac{dx^{\mu}}{d\tau}\right)\left(\frac{1}{c}\frac{dx^{\nu}}{d\tau}\right) = -1,$$

$$\Rightarrow d\tau = \frac{1}{c}\sqrt{-g_{\mu\nu}dx^{\mu}dx^{\nu}} \rightarrow \sqrt{-g_{00}}dt, \tag{37}$$

where $g_{00} < 0$. We must consider these two facts to study the cosmological phenomenon in the next sections.

At this point, it should be remarked that $\delta$ Gravity is not a traditional bigravity model. Only $g_{\mu\nu}$ is used to raise and lower indexes, the differential volume component in the action (20) just depend on $g_{\mu\nu}$ and, the most important fact, the dynamic of the universe is governed by the combination $g_{\mu\nu} + \tilde{g}_{\mu\nu}$, given by the free massless particle action in (35). This means that the phantom behaviour can be preserved even if we restrict the ghost component to turn it harmless [42–46]. Adding to quantum corrections truncated to one loop, $\delta$ Gravity could be a good model to deal with the ghost problem. On the other side, it is not a metric model of gravity too because massive particles do not move on geodesics as opposed to massless particles.

## 3. Cosmological Case

In this section, we will study photons emitted from a supernova using $\delta$ Gravity to explain the accelerated expansion of the universe without DE, with a little more detail compared with [25]. For this, we have to use the correct cosmological geometry to represent an homogeneous and isotropic universe, given by the Friedmann–Lemaître–Robertson–Walker (FLRW) metric:

$$\begin{aligned} g_{\mu\nu}dx^{\mu}dx^{\nu} &= -T^2(u)c^2du^2 + R^2(u)\left(dx^2 + dy^2 + dz^2\right), \\ \tilde{g}_{\mu\nu}dx^{\mu}dx^{\nu} &= -F_b(u)T^2(u)c^2du^2 + F_a(u)R^2(u)\left(dx^2 + dy^2 + dz^2\right), \end{aligned} \tag{38}$$

such that $T(u) = \frac{dt}{du}(u)$ and $t$ is the cosmological time. If we apply the Extended Harmonic Gauge defined in Appendix B to Equation (38), we obtain that $T(u) = T_0 R^3(u)$ and $F_b(u) = 3(F_a(u) + T_1)$, where $T_0$ and $T_1$ are gauge constants. We use $T_0 = 1$ and $T_1 = 0$ to fix the gauge completely. Thus, with these conditions, the system $(u,x,y,z)$ correspond to harmonic coordinate. Later, we can return to the usual system $(t,x,y,z)$, where $g_{\mu\nu}$ and $\tilde{g}_{\mu\nu}$ are given by:

$$g_{\mu\nu}dx^{\mu}dx^{\nu} = -c^2dt^2 + R^2(t)\left(dx^2 + dy^2 + dz^2\right), \tag{39}$$

$$\tilde{g}_{\mu\nu}dx^{\mu}dx^{\nu} = -3F_a(t)c^2dt^2 + F_a(t)R^2(t)\left(dx^2 + dy^2 + dz^2\right). \tag{40}$$

These expressions represent an isotropic and homogeneous universe. From Section 2.3, we know that the proper time is measured using the metric $g_{\mu\nu}$, but the space-time geometry is determined by the null-geodesic of $g_{\mu\nu} + \tilde{g}_{\mu\nu}$. Then, with Equations (39) and (40), we have in the Cosmological case that $t$ is the proper time and:

$$\begin{aligned} \left(g_{\mu\nu} + \tilde{g}_{\mu\nu}\right)dx^{\mu}dx^{\nu} &= 0 \\ \Rightarrow -(1 + 3F_a(t))c^2dt^2 + R^2(t)(1 + F_a(t))dr^2 &= 0. \end{aligned} \tag{41}$$

All of these will be essential considerations to explain the expansion of the universe with $\delta$ Gravity, using the supernova data.

### 3.1. Photon Trajectory and Luminosity Distance

When a photon emitted from a supernova travels to the Earth, the Universe is expanding. This means that the photon is affected by the cosmological Doppler effect. For this, we must use a null geodesic in a radial trajectory from $r_1$ to $r = 0$. Then, we can define an effective scale factor with Equation (41) as:

$$\tilde{R}(t) = R(t)\sqrt{\frac{1 + F_a(t)}{1 + 3F_a(t)}} \tag{42}$$

such that $cdt = -\tilde{R}(t)dr$. Now, if we integrate this expression from $r_1$ to $0$, we obtain:

$$r_1 = c \int_{t_1}^{t_0} \frac{dt}{\tilde{R}(t)}, \tag{43}$$

where $t_1$ and $t_0$ are the emission and reception times. If a second wave crest is emitted at $t = t_1 + \Delta t_1$ from $r = r_1$, it will reach $r = 0$ at $t = t_0 + \Delta t_0$, so:

$$r_1 = c \int_{t_1 + \Delta t_1}^{t_0 + \Delta t_0} \frac{dt}{\tilde{R}(t)}. \tag{44}$$

Therefore, if $\Delta t_0$ and $\Delta t_1$ are small, which is appropriate for light waves, we get:

$$\frac{\Delta t_0}{\Delta t_1} = \frac{\tilde{R}(t_0)}{\tilde{R}(t_1)} \Leftrightarrow \frac{\Delta \nu_1}{\Delta \nu_0} = \frac{\tilde{R}(t_0)}{\tilde{R}(t_1)}, \tag{45}$$

where $\nu_0$ is the light frequency detected at $r = 0$, corresponding to a source emission at frequency $\nu_1$. Thus, the redshift is given by:

$$1 + z(t_1) = \frac{\tilde{R}(t_0)}{\tilde{R}(t_1)}. \tag{46}$$

We see that $\tilde{R}(t)$ replaces the usual scale factor $R(t)$ to compute $z$. This means that we need to redefine the luminosity distance too. For this, let us consider a mirror of radius $b$ that receive light from our distant source at $r_1$. The photons that reach the mirror are within a cone of half-angle $\epsilon$ with origin at the source.

Let us compute $\epsilon$. The path of the light rays is given by $\vec{r}(\rho) = \rho\hat{n} + \vec{r}_1$, where $\rho > 0$ is a parameter and $\hat{n}$ is the direction of the light ray. Since the mirror is in $\vec{r} = 0$, then $\rho = r_1$ and $\hat{n} = -\hat{r}_1 + \vec{\epsilon}$, where $\epsilon$ is the angle between $-\vec{r}_1$ and $\hat{n}$ at the source, forming a cone. The proper distance is determined by the tri-dimensional metric, given by:

$$dl^2 \quad = \quad \tilde{R}^2(t)\delta_{ij}dx^i dx^j \tag{47}$$

in the cosmological case. Then, $b = \tilde{R}(t_0)r_1\epsilon$ and the solid angle of the cone is:

$$\Delta\Omega = \int_0^{2\pi} d\phi \int_0^{\epsilon} \sin(\theta)d\theta = 2\pi(1 - \cos(\epsilon)) = \pi\epsilon^2 = \frac{A}{r_1^2 \tilde{R}^2(t_0)}, \tag{48}$$

where $A = \pi b^2$ is the proper area of the mirror. Thus, the fraction of all isotropically emitted photons that reach the mirror is:

$$f = \frac{\Delta\Omega}{4\pi} = \frac{A}{4\pi r_1^2 \tilde{R}^2(t_0)}. \tag{49}$$

We know that the apparent luminosity, $l$, is the received Power per unit mirror area and Power is energy per unit time, so the received power is $P = \frac{h\nu_0}{\Delta t_0} f$, where $h\nu_0$ is the energy corresponding to the received photon. On the other side, the total emitted power by the source is $L = \frac{h\nu_1}{\Delta t_1}$, where $h\nu_1$ is the energy corresponding to the emitted photon. Therefore, we have that:

$$P = \frac{\tilde{R}^2(t_1)}{\tilde{R}^2(t_0)} L f, \tag{50}$$

$$l = \frac{P}{A} = \frac{\tilde{R}^2(t_1)}{\tilde{R}^2(t_0)} \frac{L}{4\pi r_1^2 \tilde{R}^2(t_0)}, \tag{51}$$

where we have used that $\frac{\Delta t_0}{\Delta t_1} = \frac{\nu_1}{\nu_0} = \frac{\tilde{R}(t_0)}{\tilde{R}(t_1)}$. In addition, we know that, in an Euclidean space, the luminosity decreases with distance $d_L$ according to $l = \frac{L}{4\pi d_L^2}$. Therefore, using Equation (43), the luminosity distance is:

$$d_L = \frac{\tilde{R}^2(t_0)}{\tilde{R}(t_1)} r_1 = c \frac{\tilde{R}^2(t_0)}{\tilde{R}(t_1)} \int_{t_1}^{t_0} \frac{dt}{\tilde{R}(t)}. \tag{52}$$

On the other side, we can define the angular diameter distance, $d_A$. If we consider a light ray emitted at time $t_1$ and moving in the $\theta$ coordinate, our null geodesic, given by (36), tells us that the proper distance is $s = ct_1 = \tilde{R}(t_1)r_1\theta$. The angular diameter distance is defined by $\theta = \frac{s}{d_A}$, so $d_A = \tilde{R}(t_1)r_1$. If we compare it with Equation (52), we obtain that:

$$d_A = \frac{\tilde{R}^2(t_1)}{\tilde{R}^2(t_0)} d_L = \frac{d_L}{(1+z_1)^2}. \tag{53}$$

Therefore, the relation between $d_A$ and $d_L$ is the same as in GR [49]. This result is important because, in other modified gravity theories, this relation is not satisfied [50]. We will use $d_L$ to analyze the supernovae data, but $d_A$ could be useful for other phenomena.

### 3.2. Solution of the Equations of Motion

In cosmology, the metric $g_{\mu\nu}$ is given by (39). In addition, by Equations (21) and (23), we know that Einstein's equations do not change and $T_{\mu\nu}$ is conserved. Therefore, the usual cosmological solution is still valid. Thus, using the expression in (A24) with $U_\mu = (c, 0, 0, 0)$, we obtain the well-known equations:

$$\left( \frac{\dot{R}(t)}{R(t)} \right)^2 = \frac{\kappa c^2}{3} \sum_i \rho_i(t), \qquad \dot{\rho}_i(t) = -\frac{3\dot{R}(t)}{R(t)} (\rho_i(t) + p_i(t)), \tag{54}$$

with $\dot{f}(t) = \frac{df}{dt}(t)$ and we assumed that the interaction between different components of the universe is null. To solve Equation (54), we need equations of state as $p_i(t) = \omega_i \rho_i(t)$. Since we wish to explain DE with $\delta$ Gravity, we will assume that we only have non-relativistic matter (cold DM, baryonic matter) and radiation (photons, massless particles) in the Universe. Thus, we will require two equations of state. For non-relativistic matter, we use $p_M(t) = 0$ and for radiation $p_R(t) = \frac{1}{3}\rho_R(t)$. Replacing in (54) and solving them, we find the exact solution:

$$\rho(Y) = \frac{3H_0^2 \Omega_R}{\kappa c^2 C} \frac{Y + C}{Y^4}, \tag{55}$$

$$p(Y) = \frac{H_0^2 \Omega_R}{\kappa c^2} \frac{1}{Y^4}, \tag{56}$$

$$t(Y) = \frac{2\sqrt{C}}{3H_0\sqrt{\Omega_R}} \left( \sqrt{Y + C}(Y - 2C) + 2C^{\frac{3}{2}} \right), \tag{57}$$

where $Y = \frac{R(t)}{R_0}$, $t(Y)$ is the time variable, $R_0$ is the scale factor in the present, $C = \frac{\Omega_R}{\Omega_M}$, and $\Omega_R$ and $\Omega_M = 1 - \Omega_R$ are the radiation and non-relativistic matter density in the present, respectively. We know that $\Omega_R \ll 1$, so $\Omega_M \sim 1$ and $C \ll 1$. We can see that $Y$ can be used as the independent variable. By definition, $Y \gg C$ describes the non-relativistic era and $Y \ll C$ describes the radiation era.

The equation of motion for $\tilde{g}_{\mu\nu}$ is given by (22) and (24), where $\tilde{T}_{\mu\nu}$ is a new energy-momentum tensor for $\delta$ non-relativistic matter and radiation densities, given by $\tilde{\rho}_M$ and $\tilde{\rho}_R$, respectively. Thus, using (A25) and (55)–(57), the Equations (22) and (24) are reduced to:

$$F'_a(Y) + \frac{2c\kappa}{9H_0^2\Omega_M}Y^2\left(Y\tilde{\rho}'_M(Y) + 3\tilde{\rho}_M(Y)\right) = 0, \tag{58}$$

$$F'_a(Y) + \frac{c\kappa}{6H_0^2\Omega_R}Y^3\left(Y\tilde{\rho}'_R(Y) + 4\tilde{\rho}_R(Y)\right) = 0, \tag{59}$$

$$YF'_a(Y) - 3F_a(Y) - \frac{c\kappa}{3H_0^2\Omega_M}\frac{Y^4}{(C+Y)}\left(\tilde{\rho}_M(Y) + \tilde{\rho}_R(Y)\right) = 0, \tag{60}$$

where we used $\tilde{p}_M = 0$, $\tilde{p}_R = \frac{1}{3}\tilde{\rho}_R$ and $U_\mu^T = 0$. The solution of these equations are:

$$F_a(Y) = \frac{3}{2}(2C_2 - C_1)\frac{Y}{C}\left(\sqrt{\frac{Y}{C}+1}\ln\left(\frac{\sqrt{\frac{Y}{C}+1}+1}{\sqrt{\frac{Y}{C}+1}-1}\right) - 2\right) - 2C_2 + C_3\frac{Y}{C}\sqrt{\frac{Y}{C}+1}, \tag{61}$$

$$\tilde{\rho}_M(Y) = \frac{9H_0^2\Omega_R}{2\kappa c^2 C}\frac{(C_1 - F_a(Y))}{Y^3}, \tag{62}$$

$$\tilde{\rho}_R(Y) = \frac{6H_0^2\Omega_R}{\kappa c^2}\frac{(C_2 - F_a(Y))}{Y^4}, \tag{63}$$

where $C_1$, $C_2$ and $C_3$ are integration constants. $\tilde{\rho}_M(Y)$ and $\tilde{\rho}_R(Y)$ are densities of $\delta$ Matter, so they must be not-negative functions. Then,

$$C_1 - F_a(Y) \geq 0 \quad \wedge \quad C_2 - F_a(Y) \geq 0, \tag{64}$$

for all $Y \geq 0$. Evaluating (64) at $Y = 0$, we get $C_2 \geq 0$ and $2C_2 + C_1 \geq 0$. On the other side, at $Y \gg C$, we get $C_3 \leq 0$. Now, if we use Equation (61) in (42) and define $\tilde{Y} = \frac{\tilde{R}(t)}{R(t_0)}$, we can see that:

$$\frac{\tilde{Y}}{Y} \sim \sqrt{\frac{1-2C_2}{1-6C_2}}\left(1 + \frac{3(2C_2 - C_1)}{2C(1-6C_2)(1-2C_2)}Y\log(Y)\right) + O(Y), \tag{65}$$

when $Y \ll C$. $\tilde{Y}$ is the effective scale factor, so it represents the evolution of the universe. We know that an accelerated expansion must be produced at late times, but the expansion must be driven by the non-relativistic matter and radiation at early times, this means $\frac{\tilde{Y}}{Y} = 1 + O(Y)$. For this, we have to fix $C_1 = 0$ and $C_2 = 0$ to guarantee the temporal behavior of expansion is just like GR at early times. The other constants will be chosen such that a Big-Rip is produced. That is, $\tilde{Y}(Y_{Rip}) = \infty$. We need a Big-Rip to explain the accelerated expansion of the universe because we want that $\tilde{Y}$ grows quickly when $Y$ is bigger.

The Big-Rip is determined by $C_3$, but a very small value for this parameter is necessary, otherwise the Big-Rip would be too early. However, if we use $C_3 = 0$, the Big-Rip is not produced and we cannot explain the accelerated expansion of the universe. Thus, using $C_3 = -\frac{C^{\frac{3}{2}}L_2}{3}$, with $1 \gg C \neq 0$, the effective scale factor and the Big-Rip are given by:

$$\tilde{Y}(Y, L_1, L_2, C) = Y\sqrt{\frac{1 - L_2\frac{Y}{3}\sqrt{Y+C}}{1 - L_2Y\sqrt{Y+C}}}, \tag{66}$$

$$Y_{Rip} = \left(\frac{1}{L_2}\right)^{\frac{2}{3}}. \tag{67}$$

In summary, we have that $\tilde{Y} \sim Y$ in the radiation era, where $Y \ll C$, so the Universe evolves without differences with GR. However, in some moment during the non-relativistic matter era, where $Y \gg C$, an accelerated expansion is produced, ending in a Big-Rip. We will give more details for this when we study the supernovae data. Additionally, inequalities in (64) are always satisfied; then, the $\delta$ densities are non-negative.

### 3.3. Analysis and Results

Before we start the data analysis, we must define the parameters of the model. In the first place, $d_L$ in GR depends upon four parameters: $Y$, $H_0 = 100h$ km s$^{-1}$ Mpc$^{-1}$, $\Omega_M$ and $\Omega_R$. However, from CMB black body spectrum, we obtain the photons density in the present, $\Omega_\gamma$. Now, if we assume that $\Omega_R = \Omega_\gamma + \Omega_\nu = \left(1 + 3\left(\frac{7}{8}\right)\left(\frac{4}{11}\right)^{4/3}\right)\Omega_\gamma$ ($\Omega_\nu$ is the primordial neutrino density), we get $h^2\Omega_R = 4.15 \times 10^{-5}$. Therefore, the parameters in $d_L$ can be reduced to three: $Y$, $h$ and $h^2\Omega_M$. In the same way, in $\delta$ Gravity with $\delta$ Matter, $d_L$ depends on three parameters: $Y$, $C$ and $L_2$. We will use $H_0\sqrt{\Omega_R} = 0.644$ km s$^{-1}$ Mpc$^{-1}$.

The supernovae data give the apparent magnitude, $m$, as a function of redshift, $z$. For this reason, it is useful to use $z$ instead of $Y$. The apparent magnitude is:

$$m(z) = M + 5\log_{10}\left(\frac{d_L(z)}{10 \text{ pc}}\right), \tag{68}$$

where $M$ is the absolute magnitude, constant and common for all supernovae.

Finally, the difference between GR and $\delta$ Gravity will be given by $d_L(z)$. Thus, the luminosity distance expressions are:

$$\textbf{GR:} \quad d_L(z) = \frac{c(1+z) \text{ Mpc s}}{100 \text{ km}} \int_{\frac{1}{1+z}}^{1} \frac{dY'}{\sqrt{h^2\left(1 - \Omega_M - \Omega_R\right)Y'^4 + h^2\Omega_M Y' + h^2\Omega_R}}, \tag{69}$$

$$\boldsymbol{\delta} \textbf{ Gravity:} \quad d_L(z) = c(1+z)\frac{\sqrt{C}}{H_0\sqrt{\Omega_R}} \int_0^z \frac{(1+u)Y(u)\frac{dY}{du}(u)}{\sqrt{Y(u) + C}}du, \tag{70}$$

where we used Equations (52) and (57) for $\delta$ Gravity. Besides, $Y(z)$ must satisfy $\tilde{Y}(Y(z)) = \frac{\tilde{Y}_0}{1+z}$ in (70), with $\tilde{Y}_0 = \tilde{Y}(1)$ and $\tilde{Y}(Y(z))$ given by Equation (66).

The statistical method used to interpret errors in data is given by the variance $\sigma$ in a normally distributed random variable. In our case, the data is given by $(z_i, \mu_i)$, where:

$$\mu(z) \equiv m(z) - M = 5\log_{10}\left(\frac{d_L(z)}{10 \text{ pc}}\right)$$

is the distance modulus. Then, we must minimize:

$$\chi^2(\text{per point}) = \frac{1}{N}\sum_{i=1}^{N} \frac{(\mu_i - \mu(z_i))^2}{\sigma_{\mu_i}^2}, \tag{71}$$

where $N$ is the number of data points and $\sigma_{\mu_i}$ is the error of $\mu_i$. Now, we can proceed to analyze the data given in [51] with $N = 580$ supernovae. In both cases, GR and $\delta$ Gravity, $d_L$ is given by an exact expression, but we need to use a numerical method to solve the integral and fit the data to determinate the optimum values for the parameters that represent the $\mu$ v/s $z$ of the supernovae data[1]. The parameters that minimize Equation (71) are:

- GR: $h = 0.7 \pm 3.37 \times 10^{-3}$ and $h^2\Omega_M = 0.136 \pm 8.5 \times 10^{-3}$ with $\chi^2(\text{per point}) = 0.985$,
- $\delta$ Gravity with $\delta$ Matter: $L_2 = 0.457 \pm 0.0114$ and $C = 1.89 \times 10^{-4} \pm 4.92 \times 10^{-6}$ with $\chi^2(\text{per point}) = 0.985$.

We can see in Figure 1 that $\delta$ Gravity with $\delta$ Matter fit the data very well. Now, with these values, we can compute the age of the universe and the Big-Rip era. For GR, the age of the universe is $1.377 \times 10^{10}$ years. However, in our model, the time is given by (57). Thus, substituting the corresponding values for $L_2$, $C$ and taking $Y = \frac{R(t)}{R(t_0)} = 1$, we obtain $1.391 \times 10^{10}$ years for the age of the universe. To compute when the Big-Rip will happen, we need to use Equation (67). That is, $Y_{Rip} = 1.684$, so $t_{\text{Big-Rip}} = 3.042 \times 10^{10}$ years. Therefore, the Universe has lived less than half of its life.

On the other side, in [24], we obtained that, in $\delta$ Gravity without $\delta$ Matter, the age of the universe is $1.92 \times 10^{10}$ years and $t_{\text{Big-Rip}} = 4.3 \times 10^{10}$ years. The problem in this case is the huge age of the universe, compared with Planck Collaboration given by $1.381 \times 10^{10}$ years[2]. However, we cannot say that this case is totally rejected yet, but the age of the universe for $\delta$ Gravity with $\delta$ Matter is more similar to Planck.

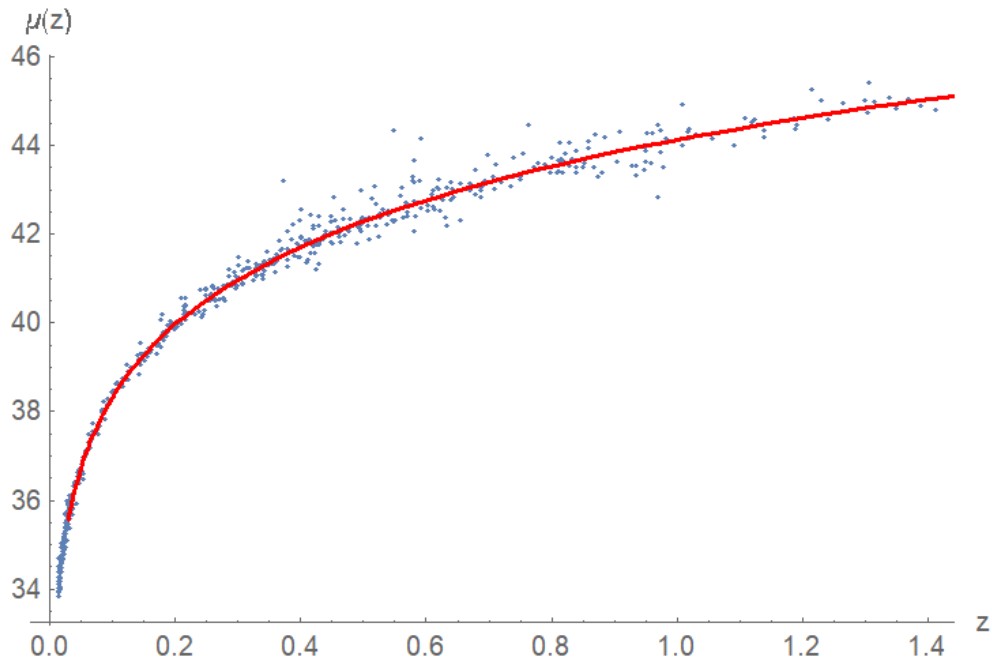

**Figure 1.** Distance modulus vs. Redshift. We have fitted 580 supernovae to $\delta$ Gravity, using the parameters that minimize Equation (71).

Now, using Equations (62) and (63), the $\delta$ Matter in the present is given by:

$$\tilde{\Omega}_M = \frac{\kappa c^2 \tilde{\rho}_M(1)}{3H_0^2} \approx \frac{L_2}{2}\Omega_M = 0.23\Omega_M \quad \wedge \quad \tilde{\Omega}_R = \frac{\kappa c^2 \tilde{\rho}_R(1)}{3H_0^2} \approx \frac{2L_2}{3}\Omega_R = 0.3\Omega_R, \tag{72}$$

---

[1]  To obtain the best combination of parameters, we used **NonLinearModelFit** from Mathematica 11.0. Then, we used these parameters to minimize Equation (71). For more details, see the Mathematica 11.0 help.

[2]  The age of the universe of Planck was calculated using the cosmological parameters obtained in [52]. That is, $\Omega_M = 0.308$ and $H_0 \equiv 100h = 67.8$ km/s/Mpc.

where $\tilde{\Omega}_M$ and $\tilde{\Omega}_R$ are the normalized $\delta$ densities in the present. Therefore, we have two components of $\delta$ Matter, related to ordinary matter at cosmological level. These components can be considered like a contribution to DM; however, a more accuracy analysis in a field theory level is necessary to understand the nature of $\delta$ Matter. In the next section, we will study the Non-Relativistic limit to understand the phenomenological effect $\delta$ Matter in the Dark Sector. In any case, $\tilde{\Omega}_M$ and $\tilde{\Omega}_R$ are important to explain the dynamic of the expansion of the universe.

### 3.4. Introduction to $\delta$ Inflation

To finish with the cosmological case, we will make a few comments about the general equations of motion for only one fluid in $\delta$ Gravity for future references. Using the cosmological solution for $g_{\mu\nu}$ and $\tilde{g}_{\mu\nu}$, given by (39) and (40) respectively, we obtain:

$$\kappa\rho(t) = 3H^2(t), \tag{73}$$

$$\kappa p(t) = -2\dot{H}(t) - 3H^2(t), \tag{74}$$

$$\kappa\tilde{\rho}(t) = 3H(t)\left(\dot{F}_a(t) - 3H(t)F_a(t)\right), \tag{75}$$

$$\kappa\tilde{p}(t) = -\ddot{F}_a(t) + 3\left(2\dot{H}(t) + 3H^2(t)\right)F_a(t), \tag{76}$$

where $H(t) = \frac{\dot{R}(t)}{R(t)}$ is the Hubble parameter and $\tilde{p}(t) = \tilde{\delta}p(t)$. To complete the system, we need equations of state to solve them. They are $p(t) = \omega(t)\rho(t)$ and $\tilde{p}(t) = \omega(t)\tilde{\rho}(t) + \tilde{\omega}(t)\rho(t)$. In a perfect fluid, $\omega(t)$ usually is assumed to be constant and $\tilde{\omega}(t)$ must be zero in that case. Thus, using these equations of state in (73)–(76), we obtain:

$$\dot{H}(t) + \frac{3}{2}\left(\omega(t) + 1\right)H^2(t) = 0, \tag{77}$$

$$\ddot{F}_a(t) + 3\omega(t)H(t)\dot{F}_a(t) = -3\tilde{\omega}(t)H^2(t). \tag{78}$$

In order to understand the behavior of these equations, we will solve the case where $\omega(t)$ is almost constant, to be close to the usual perfect fluid. The solution is:

$$R(t) = \begin{cases} R_0\left(t + t_0\right)^{\frac{2}{3(1+\omega)}}, & \text{with: } \omega \neq -1, \\ R_1 e^{\lambda t}, & \text{with: } \omega = -1, \end{cases}$$

$$F_a(t) = \begin{cases} a_0\left(\frac{R(t)}{R_2}\right)^{\frac{3}{2}(1-\omega)} + \frac{2\tilde{\omega}}{1-\omega}\ln\left(\frac{R(t)}{R_2}\right), & \text{with: } \omega \neq 1, \\ a_1\ln\left(\frac{R(t)}{R_3}\right) - \frac{3\tilde{\omega}}{2}\ln^2\left(\frac{R(t)}{R_3}\right), & \text{with: } \omega = 1, \end{cases} \tag{79}$$

where $R_0$, $R_1$, $R_2$, $R_3$, $t_0$, $a_0$ and $a_1$ are integration constants. From this result, we notice that $R(t)$ obey the standard power-law solution in a perfect fluid with a constant $\omega$. However, we must remember that the dynamic of the universe in $\delta$ Gravity is given by the effective scale factor, (42), which produce an accelerated expansion. This means that we can define an effective Hubble parameter given by $\mathcal{H}(t) \equiv \frac{\dot{\tilde{R}}(t)}{\tilde{R}(t)}$. In this case, with $\tilde{\omega} = 0$, it is:

$$\mathcal{H}(X) = \begin{cases} H(X)\left(1 + \dfrac{3c_2(1-\omega)X^{\frac{3}{2}(1-\omega)}}{2\left(c_1 + 2c_2 - X^{\frac{3}{2}(1-\omega)}\right)\left(c_1 - X^{\frac{3}{2}(1-\omega)}\right)}\right), & \text{with: } \omega \neq 1, \\[20pt] H(X)\left(1 + \dfrac{d_2}{(d_1 + 2d_2 - \ln(X))(d_1 - \ln(X))}\right), & \text{with: } \omega = 1, \end{cases} \tag{80}$$

where $X = \frac{R(t)}{R_0}$ and $c_1$, $c_2$, $d_1$ and $d_2$ are integration constants. Therefore, even with a standard power-law solution, we can obtain a different behavior. However, we have to say that the power-law solution is just for a perfect fluid. Actually, the solution of $R(t)$ in some specific model will be the same solution obtained in GR. This is because the Einstein's equations are preserved in $\delta$ Gravity, but the dynamic is affected by the effective scale factor. For example, in inflation, a scalar field is used to produce the exponential expansion. In that case:

$$
\begin{aligned}
\rho(t) &= \frac{1}{2}\dot{\varphi}_0^2(t) + V(\varphi_0(t)), \\
p(t) &= \frac{1}{2}\dot{\varphi}_0^2(t) - V(\varphi_0(t)), \\
\tilde{\rho}(t) &= \dot{\varphi}_0(t)\dot{\tilde{\varphi}}_0(t) - \frac{3}{2}F_a(t)\dot{\varphi}_0^2(t) + V_{,\varphi}(\varphi_0(t))\tilde{\varphi}_0(t), \\
\tilde{p}(t) &= \dot{\varphi}_0(t)\dot{\tilde{\varphi}}_0(t) - \frac{3}{2}F_a(t)\dot{\varphi}_0^2(t) - V_{,\varphi}(\varphi_0(t))\tilde{\varphi}_0(t).
\end{aligned}
$$

With GR, inflation must obey $V(\varphi_0(t)) \gg \dot{\varphi}_0^2(t)$ to obtain $\omega(t) = \frac{p(t)}{\rho(t)} \sim -1$, such that the expansion is exponential (see Equation (79)). However, in $\delta$ Gravity, the accelerate expansion could be produced by a divergence in $\tilde{R}(t)$, just like we explained DE. Additionally, in inflation, we have a new field, $\tilde{\varphi}_0$, giving us a non-zero $\tilde{\omega}(t)$. In conclusion, in inflation with $\delta$ Gravity, an accelerated expansion can be produced by additional factors. Basically, the expansion rate is governed by $\mathcal{H}(t) = \frac{\dot{\tilde{R}}(t)}{\tilde{R}(t)}$. Now, an *effective $\omega$ parameter* can be defined as:

$$
\omega_{eff}(t) = -\frac{2\dot{\mathcal{H}}(t)}{3\mathcal{H}^2(t)} - 1. \tag{81}
$$

If we apply Equation (80) in (81), we can study the expansion behaviour of our model. If $\omega_{eff}(t) < -1$, the expansion is like a phantom model. This calculation is briefly developed in [53], where we demonstrated that $\delta$ Gravity works like a phantom model. In addition, Equation (81) can be used to compare other alternative theories with $\delta$ Gravity in a cosmological level. A more detailed version of this work is in progress.

## 4. Non-Relativistic Case

In this section, we will consider the Newtonian and Post-Newtonian limit to study new effects on $\delta$ Gravity. We expect a weak deviation of GR at solar system scale, but we want to see in a galactic scale if DM could be explained with $\delta$ Matter.

### 4.1. Newtonian and Post-Newtonian Limit

If we introduce one order more for the Newtonian limit, the metric will be given by [54]:

$$
\begin{aligned}
g_{\mu\nu}dx^\mu dx^\nu &= -\left(1 + 2\phi\epsilon^2 + 2\left(\phi^2 + \psi\right)\epsilon^4\right)\left(\frac{cdt}{\epsilon}\right)^2 + \left(1 - 2\phi\epsilon^2 - 2\psi\epsilon^4\right)\left(dx^2 + dy^2 + dz^2\right) \\
&\quad + 2\epsilon^3\left(\chi_1 dx + \chi_2 dy + \chi_3 dz\right)\left(\frac{cdt}{\epsilon}\right) + \epsilon^4\left(\xi_{11}dx^2 + \xi_{22}dy^2 + \xi_{33}dz^2\right. \\
&\quad \left. + 2\xi_{12}dxdy + 2\xi_{13}dxdz + 2\xi_{23}dydz\right),
\end{aligned} \tag{82}
$$

where $\epsilon \sim \frac{v}{c}$ is the perturbative parameter and $g_{\mu\nu} \to \eta_{\mu\nu}$ for $r \to \infty$. We can see the Newtonian limit represented by $\phi$ at order $\epsilon^2$ in the components of $g_{\mu\nu}$. In the Post-Newtonian limit, we have ten additional functions to represent ten degrees of freedom on the metric. In the same way, $\tilde{g}_{\mu\nu}$ will be:

$$\tilde{g}_{\mu\nu}dx^{\mu}dx^{\nu} = -2\left(\tilde{\phi}\epsilon^2 + (2\phi\tilde{\phi} + \tilde{\psi})\epsilon^4\right)\left(\frac{cdt}{\epsilon}\right)^2 - 2\left(\tilde{\phi}\epsilon^2 + \tilde{\psi}\epsilon^4\right)\left(dx^2 + dy^2 + dz^2\right)$$

$$+2\epsilon^3\left(\tilde{\chi}_1 dx + \tilde{\chi}_2 dy + \tilde{\chi}_3 dz\right)\left(\frac{cdt}{\epsilon}\right) + \epsilon^4\left(\tilde{\xi}_{11}dx^2 + \tilde{\xi}_{22}dy^2 + \tilde{\xi}_{33}dz^2\right) \quad (83)$$

$$+2\tilde{\xi}_{12}dxdy + 2\tilde{\xi}_{13}dxdz + 2\tilde{\xi}_{23}dydz\Big),$$

where we used $\tilde{g}_{\mu\nu} \to 0$ for $r \to \infty$. All functions in Equations (82) and (83) depend on $(t, x, y, z)$, but $\frac{1}{c}\frac{\partial}{\partial t} \sim \epsilon$. For this reason, we use $ct \to \frac{ct}{\epsilon}$ to obtain the equations. Equations (82) and (83) are the more general expression for a covariant tensor of rank two, so we need to fix a gauge. One particularly convenient gauge is given by the extended harmonic coordinate conditions presented in Appendix B. Using Equations (A26) and (A27), we obtain:

$$4\dot{\phi} + \partial_i\chi_i = 0, \quad (84)$$

$$2\phi\partial_i\phi - \dot{\chi}_i - \frac{1}{2}\partial_i\xi_{jj} + \partial_j\xi_{ij} = 0, \quad (85)$$

$$4\dot{\tilde{\phi}} + \partial_i\tilde{\chi}_i = 0, \quad (86)$$

$$2\phi\partial_i\tilde{\phi} + 2\tilde{\phi}\partial_i\phi - \dot{\tilde{\chi}}_i - \frac{1}{2}\partial_i\tilde{\xi}_{jj} + \partial_j\tilde{\xi}_{ij} = 0. \quad (87)$$

On the other side, in Appendix A, it is proved that the energy-momentum tensors for a perfect fluid are given by (A24) and (A25). Now, in the Non-Relativistic limit, we have that:

$$\rho = \rho^{(0)} + \epsilon^2\rho^{(2)}, \quad (88)$$

$$\tilde{\rho} = \tilde{\rho}^{(0)} + \epsilon^2\tilde{\rho}^{(2)}, \quad (89)$$

$$p(\rho) = \epsilon^2 p^{(2)}(\rho), \quad (90)$$

$$U_\mu = \left(c\left(\frac{1 + \epsilon^2\left(\phi + \frac{1}{2}U_k^{(1)}U_k^{(1)}\right)}{\epsilon}\right), \epsilon U_i^{(1)}\right), \quad (91)$$

$$U_\mu^T = \left(c\epsilon U_k^{T(1)}U_k^{(1)}, \epsilon U_i^{T(1)}\right), \quad (92)$$

reducing the equations of motion (21) and (22) to:

$$\partial^2\phi = \frac{\kappa}{2}\rho^{(0)}, \quad (93)$$

$$\partial^2\chi_i = -2\kappa U_i^{(1)}\rho^{(0)}, \quad (94)$$

$$\partial^2\psi = \frac{\kappa}{2}\left(2\left(U_k^{(1)}U_k^{(1)} - \phi\right)\rho^{(0)} + \rho^{(2)} + 3p^{(2)}(\rho)\right) + \ddot{\phi}, \quad (95)$$

$$\partial^2\xi_{ij} = -2\kappa U_i^{(1)}U_j^{(1)}\rho^{(0)} - 4(\partial_i\phi)(\partial_j\phi) + 2\kappa\left(\left(U_k^{(1)}U_k^{(1)} + \phi\right)\rho^{(0)} + 2p^{(2)}(\rho)\right)\delta_{ij}$$

$$+4(\partial_k\phi)(\partial_k\phi)\delta_{ij}, \quad (96)$$

$$\partial^2\tilde{\phi} = \frac{\kappa}{2}\tilde{\rho}^{(0)}, \quad (97)$$

$$\partial^2\tilde{\chi}_i = -2\kappa\left(U_i^{T(1)}\rho^{(0)} + U_i^{(1)}\tilde{\rho}^{(0)}\right), \quad (98)$$

$$(99)$$

$$\partial^2 \tilde{\psi} = \kappa \left( \left( 2 U_k^{(1)} U_k^{T(1)} - \tilde{\phi} \right) \rho^{(0)} + \left( U_k^{(1)} U_k^{(1)} - \phi + \frac{3}{2} p'^{(2)}(\rho) \right) \tilde{\rho}^{(0)} + \frac{\tilde{\rho}^{(2)}}{2} \right) + \ddot{\tilde{\phi}}, \tag{100}$$

$$\begin{aligned}
\partial^2 \tilde{\xi}_{ij} = &-2\kappa \left( \left( U_i^{T(1)} U_j^{(1)} + U_i^{(1)} U_j^{T(1)} \right) \rho^{(0)} + U_i^{(1)} U_j^{(1)} \tilde{\rho}^{(0)} \right) - 4(\partial_i \tilde{\phi})(\partial_j \phi) - 4(\partial_i \phi)(\partial_j \tilde{\phi}) \\
&+2\kappa \left( \left( 2 U_k^{(1)} U_k^{T(1)} + \tilde{\phi} \right) \rho^{(0)} + \left( U_k^{(1)} U_k^{(1)} + \phi + 2 p'^{(2)}(\rho) \right) \tilde{\rho}^{(0)} \right) \delta_{ij} \\
&+8(\partial_k \phi)(\partial_k \tilde{\phi}) \delta_{ij},
\end{aligned} \tag{101}$$

where $p'^{(2)}(\rho) = \frac{\partial p^{(2)}}{\partial \rho}(\rho)$, $\partial^2 = \partial_i \partial_i$, $\partial_i = \frac{\partial}{\partial x^i}$ and $\kappa \sim O(\epsilon^2)$. We can see that Equations (93) and (97) correspond to the Newtonian limit. To complete the system, we have Equations (23) and (24), but they are automatically satisfied when we consider (84)–(87). However, they are useful because we can write them in terms of $\rho^{(0)}$, $\rho^{(2)}$, $\tilde{\rho}^{(0)}$, $\tilde{\rho}^{(2)}$ and $p^{(2)}$. In spherical symmetry with $U_i^{(1)} = U_i^{T(1)} = 0$, it is possible to prove that the conservation equations can be reduced to:

$$\frac{\partial p^{(2)}}{\partial r}(r) + \rho^{(0)}(r) \left( \frac{\partial \phi}{\partial r}(r) \right) = 0, \tag{102}$$

$$\frac{\partial}{\partial r} \left( \left( 1 + 2\epsilon^2 \phi(r) \right)^{-1} \left( \tilde{\phi}(r) - \tilde{\rho}^{(0)}(r) \frac{\left( \frac{\partial \phi}{\partial r}(r) \right)}{\left( \frac{\partial \rho^{(0)}}{\partial r}(r) \right)} \right) \right) = 0. \tag{103}$$

From (102), we can obtain $p^{(2)}(r)$ and Equation (103) say us that:

$$\tilde{\rho}^{(0)}(r) = \frac{\left( \frac{\partial \rho^{(0)}}{\partial r}(r) \right)}{\left( \frac{\partial \phi}{\partial r}(r) \right)} \left( \tilde{\phi}(r) - C \left( 1 + 2\epsilon^2 \phi(r) \right) \right), \tag{104}$$

where $C$ is an integration constant. We preserve $2\epsilon^2 \phi(r)$ in the last term of the right side because the order of $C$ is unknown. Equations (102) and (103) are obtained in a Post-Newtonian level. This means that we need to study the system at this level to obtain the relation (104) and complete the Newtonian limit. Finally, Equations (93) and (97) for spherical symmetry are reduced to:

$$\frac{1}{r^2} \frac{\partial}{\partial r} \left( r^2 \left( \frac{\partial \phi(r)}{\partial r} \right) \right) = \frac{\kappa}{2} \rho^{(0)}(r), \tag{105}$$

$$\frac{1}{r^2} \frac{\partial}{\partial r} \left( r^2 \left( \frac{\partial \tilde{\phi}(r)}{\partial r} \right) \right) = \frac{\kappa}{2} \frac{\left( \frac{\partial \rho^{(0)}}{\partial r}(r) \right)}{\left( \frac{\partial \phi}{\partial r}(r) \right)} \left( \tilde{\phi}(r) - C \left( 1 + 2\epsilon^2 \phi(r) \right) \right). \tag{106}$$

Then, to obtain $\phi(r)$ and $\tilde{\phi}(r)$, we just need $\rho^{(0)}(r)$, completing the Newtonian limit. Now, we can ask ourselves if it is possible to explain DM with this result. For this, we will need to study the trajectory of a free particle.

### 4.2. Trajectory of a Particle:

From Section 2.3, we know that the acceleration is given by (30). In the Post-Newtonian limit, it is reduced to [54]:

$$\begin{aligned}
\frac{1}{c^2} \frac{d^2 \vec{x}}{dt^2} = &-\epsilon^2 \nabla \left( \phi_N + \left( 2\phi_N^2 + \psi_N \right) \epsilon^2 \right) \\
&+\epsilon^4 \left( 3\vec{v}\dot{\phi}_N + 4\vec{v} \left( \vec{v} \cdot \nabla \phi_N \right) - v^2 \nabla \phi_N - \dot{\vec{\chi}}_N + \left( \vec{v} \times \nabla \times \vec{\chi}_N \right) \right) \\
&+\frac{\epsilon^4}{2} \nabla \tilde{\phi}^2 + O\left( \epsilon^6 \right),
\end{aligned} \tag{107}$$

where $\vec{v} = \frac{d\vec{x}}{dt}$, $\phi_N = \phi + \tilde{\phi}$ and analogous expressions for the others fields. From (107), we can deduce a couple of things. Firstly, in the Newtonian limit, we have:

$$\frac{1}{c^2}\frac{d^2\vec{x}}{dt^2} = -\epsilon^2 \nabla \phi_N, \tag{108}$$

so $\phi_N$ is the effective Newtonian potential. Secondly, the acceleration is similar to the usual case if we replace $\phi \to \phi_N$, with the exception of the last term in (107). If we analyze the case with spherical symmetry far away from matter, we can see from Equation (106) that $\tilde{\phi}^2 \sim r^{-2}$. This means that this term is $\sim -r^{-3}$, therefore it is an attractive contribution and can be considered as a contribution to DM. In this paper, we will focus on the Newtonian approximation, given by (108), which is the dominant term. The contribution by other terms will be considered in future works.

We have said that $\phi_N$ is the effective potential in the Newtonian limit. This means that the effective density is $\rho_{eff} = \rho^{(0)} + \tilde{\rho}^{(0)}$. In spherical symmetry, that is:

$$\rho_{eff}(r) = \rho^{(0)}(r) + \frac{\left(\frac{\partial \rho^{(0)}}{\partial r}(r)\right)}{\left(\frac{\partial \phi}{\partial r}(r)\right)}\left(\tilde{\phi}(r) - C\left(1 + 2\epsilon^2\phi(r)\right)\right). \tag{109}$$

Therefore, the second term in (109) is an additional mass and it could be identified as DM. To verify this, we will study some density profile used to fit the galaxy distribution and then obtain the effective density. Next, we will analyze Equation (109) and the equations of motion of $\phi$ and $\tilde{\phi}$ to see if $\delta$ Matter can explain the DM effect.

*4.3. Density Profiles*

To study the $\delta$ Matter effects, we must analyze Equations (104)–(106). To explain DM, $\delta$ Matter must be negligible in the solar system scale, but important in galactic scale. We will study these equations with some density profile. In the first place, we will see a spherically homogeneous density like a first approximation for a planetary or stellar distribution. Then, we will study the exponential profile and finally we will use the Einasto and Navarro–Frenk–White profiles to describe galaxies' distributions. We define a normalized radius $x$ such that $r = Rx$; then, our equations are:

$$\frac{1}{x^2}\frac{\partial}{\partial x}\left(x^2\left(\frac{\partial \phi(x)}{\partial x}\right)\right) = \frac{\kappa R^2}{2}\rho(x), \tag{110}$$

$$\frac{1}{x^2}\frac{\partial}{\partial x}\left(x^2\left(\frac{\partial \tilde{\phi}(x)}{\partial x}\right)\right) = \frac{\kappa R^2}{2}\frac{\left(\frac{\partial \rho}{\partial x}(x)\right)}{\left(\frac{\partial \phi}{\partial x}(x)\right)}\left(\tilde{\phi}(x) - C\left(1 + 2\epsilon^2\phi(x)\right)\right), \tag{111}$$

where $R$ is a convenient radius. With these equations, we can define the ordinary mass and tilde mass. That is:

$$m(x) = 4\pi R^3 \int_0^\infty dx\, x^2 \rho(x) = \frac{8\pi R}{\kappa}x^2\left(\frac{\partial \phi(x)}{\partial x}\right), \tag{112}$$

$$\tilde{m}(x) = 4\pi R^3 \int_0^\infty dx\, x^2 \tilde{\rho}(x) = \frac{8\pi R}{\kappa}x^2\left(\frac{\partial \tilde{\phi}(x)}{\partial x}\right), \tag{113}$$

such that the effective mass is $M(x) = m(x) + \tilde{m}(x)$. Finally, from Equation (108), we have that the rotation velocity is:

$$\left(\frac{v_{rot}(x)}{c}\right)^2 = \epsilon^2 x \frac{\partial}{\partial x}\left(\phi(x) + \tilde{\phi}(x)\right) = \frac{\epsilon^2 \kappa M(x)}{8\pi R x}, \tag{114}$$

where we used that $\frac{d^2\vec{x}}{dt^2} = -\frac{v_{rot}^2(r)}{r}\hat{\theta}$ in this case.

### 4.3.1. Spherically Homogeneous Profile

We can think, in a first approximation, that planets or stars are spheres with a constant density. That is, $\rho(x) = \rho_0\Theta(1-x)$. Thus, Equations (110) and (111) are:

$$\frac{1}{x^2}\frac{\partial}{\partial x}\left(x^2\left(\frac{\partial\phi(x)}{\partial x}\right)\right) = \frac{\kappa R^2 \rho_0}{2}\Theta(1-x)$$

$$\frac{1}{x^2}\frac{\partial}{\partial x}\left(x^2\left(\frac{\partial\tilde{\phi}(x)}{\partial x}\right)\right) = -\frac{\kappa R^2 \rho_0}{2}\frac{\delta(1-x)}{\left(\frac{\partial\phi}{\partial x}(x)\right)}\left(\tilde{\phi}(x) - C\left(1 + 2\epsilon^2\phi(x)\right)\right)$$

$$\tilde{\rho}(x) = -\rho_0 \frac{\delta(1-x)}{\left(\frac{\partial\phi}{\partial x}(x)\right)}\left(\tilde{\phi}(x) - C\left(1 + 2\epsilon^2\phi(x)\right)\right),$$

where $R$ is the radius of the sphere. From the first equation, we can obtain $\phi(x)$. Using the boundary conditions $\phi(\infty) \to 0$, $\phi(0) \to$ *"finity value"* and imposing that $\phi(x)$ and $\phi'(x)$ must be continuous for all $x$, we obtain:

$$\phi(x) = \begin{cases} \frac{\kappa R^2 \rho_0}{12}\left(x^2 - 3\right) & x \leq 1, \\ -\frac{\kappa R^2 \rho_0}{6x} & x > 1. \end{cases} \tag{115}$$

On the other side, from the second equation, we obtain $\tilde{\phi}(x)$, but we can not impose a continuous $\tilde{\phi}'(x)$. Instead, the equation of motion of $\tilde{\phi}(x)$ tells us:

$$x^2\left(\frac{\partial\tilde{\phi}(x)}{\partial x}\right)\Big|_{int}^{ext} = -\frac{\kappa R^2 \rho_0}{2\phi'(1)}\left(\tilde{\phi}(1) - C\left(1 + 2\epsilon^2\phi(1)\right)\right),$$

where the other conditions on $\tilde{\phi}(x)$ are the same. They are $\tilde{\phi}(\infty) \to 0$, $\tilde{\phi}(0) \to$ *"finity value"* and $\tilde{\phi}(x)$ is continuous for all $x$. Then, the solution is:

$$\tilde{\phi}(x) = \begin{cases} \frac{3}{2}C\left(1 - \frac{\epsilon^2\kappa R^2\rho_0}{3}\right) & x \leq 1, \\ \frac{3}{2x}C\left(1 - \frac{\epsilon^2\kappa R^2\rho_0}{3}\right) & x > 1. \end{cases} \tag{116}$$

In addition, the $\delta$ Matter density is:

$$\tilde{\rho}(x) = -\frac{3C}{\kappa R^2}\left(1 - \frac{\epsilon^2\kappa R^2\rho_0}{3}\right)\delta(x-1) \approx -\frac{3C}{\kappa R^2}\delta(x-1) \tag{117}$$

and the acceleration is given by (108), then:

$$\vec{a} = -\epsilon^2\frac{c^2}{R}\frac{\partial}{\partial x}\left(\phi(x) + \tilde{\phi}(x)\right)\hat{r}. \tag{118}$$

Naturally, we expect a continuous $\vec{a}$, but we saw that $\tilde{\phi}'(x)$ is not. This means that we have to accept that $\delta$ Gravity produces an additional force on the surface of the sphere, or $C = 0$. In the last case, all $\delta$ components disappear, so $\delta$ Gravity is the same as GR. This result is a really important condition because $\delta$ Matter will be negligible when the distribution of ordinary matter can be represented by a homogeneous sphere. This case can be an acceptable representation of planets and stars, where $\delta$ Matter does not produce important effects. However, $\delta$ Matter could be important when the distribution of ordinary matter is not homogeneous like a galaxy, where the DM effects are important.

### 4.3.2. Exponential Profile

We will study this profile to develop an acceptable first approximation to a galaxy distribution. That is, $\rho(x) = \rho_0 e^{-x}$. In this case, Equations (110) and (111) are:

$$\frac{1}{x^2}\frac{\partial}{\partial x}\left(x^2\left(\frac{\partial \phi(x)}{\partial x}\right)\right) = \frac{\kappa R^2 \rho_0 e^{-x}}{2},$$

$$\frac{1}{x^2}\frac{\partial}{\partial x}\left(x^2\left(\frac{\partial \tilde{\phi}(x)}{\partial x}\right)\right) = -\frac{\kappa R^2 \rho_0 e^{-x}}{2\left(\frac{\partial \phi}{\partial x}(x)\right)}\left(\tilde{\phi}(x) - C\left(1 + 2\epsilon^2 \phi(x)\right)\right).$$

From the first equation, we can solve $\phi(x)$ analytically. Using the boundary conditions $\phi(\infty) \to 0$ and $\phi(0) \to$ *"finity value"*, we obtain:

$$\phi(x) = \frac{\kappa R^2 \rho_0 \left((x+2)e^{-x} - 2\right)}{2x}. \tag{119}$$

On the other side, the equation of $\tilde{\phi}(x)$ is too complicated, which is:

$$\frac{\partial}{\partial x}\left(x^2\left(\frac{\partial \tilde{\phi}(x)}{\partial x}\right)\right) = -\frac{x^3 e^{-x}\left(x\tilde{\phi}(x) - C\left(x - \epsilon^2 \kappa R^2 \rho_0\left(2 - (x+2)e^{-x}\right)\right)\right)}{2 - (x^2 + 2x + 2)e^{-x}}. \tag{120}$$

Now, we can solve this equation numerically and find $m(x)$ and $\tilde{m}(x)$. First, we must analyze the initial conditions. We can do it studying the behavior of $m(x)$ and $\tilde{m}(x)$, given by (112) and (113), for $x \ll 1$ because they are related to $x^2\left(\frac{\partial \phi(x)}{\partial x}\right)$ and $x^2\left(\frac{\partial \tilde{\phi}(x)}{\partial x}\right)$, respectively. From the solution (119), we have that $\phi(x) \approx \frac{\kappa R^2 \rho_0}{2}$ and $m(x) \approx \frac{4\pi R^3 \rho_0}{3}x^3$ for small $x$. On the other side, from Equation (120), we need $\tilde{\phi}(x) \approx C\left(1 - \epsilon^2 \kappa R^2 \rho_0\right)$ and $\tilde{m}(x) \approx \epsilon^2 C\pi R^3 \rho_0 x^4$. Thus, the total mass is completely dominated by ordinary components in the center.

Previously, we said that the $C$ order is unknown, but the initial conditions say us that $\frac{\tilde{m}(x)}{m(x)} \approx \frac{3\epsilon^2 C}{4}x$. This means that, the $\delta$ Matter is increased from $x << 1$ too slowly, unless $\epsilon^2 C \sim O(1)$. We can see this in Figure 2a, where we present to $m(x)$ and $\tilde{m}(x)$. Additionally, we represent the relation between $m(x)$ and $\tilde{m}(x)$ in Figure 2b. These plots say us that ordinary matter is dominant in the center, but $\delta$ Matter is accumulated when we get away from the center and the behavior of both kinds of matters become similar when $x$ increase. In fact, the relation between them is practically constant in the edge of the galaxy, $\frac{\tilde{m}(x)}{m(x)} \to 3\epsilon^2 C$. In conclusion, ordinary and $\delta$ Matter evolve in a similar way, but $\delta$ Matter is concentrated outside of the galactic nucleus.

In Figure 2c, we show the rotation velocity in $\delta$ Gravity with $\epsilon^2 C = \left\{\frac{1}{2}, 1, \frac{3}{2}\right\}$ and GR. Clearly, the similar behavior of both elements and the additional mass from $\delta$ Matter produce an amplifying effect in the rotation velocity. This means that a minimal quantity of ordinary matter could explain the rotation velocity in a galaxy because of the additional contribution produced by $\delta$ Matter.

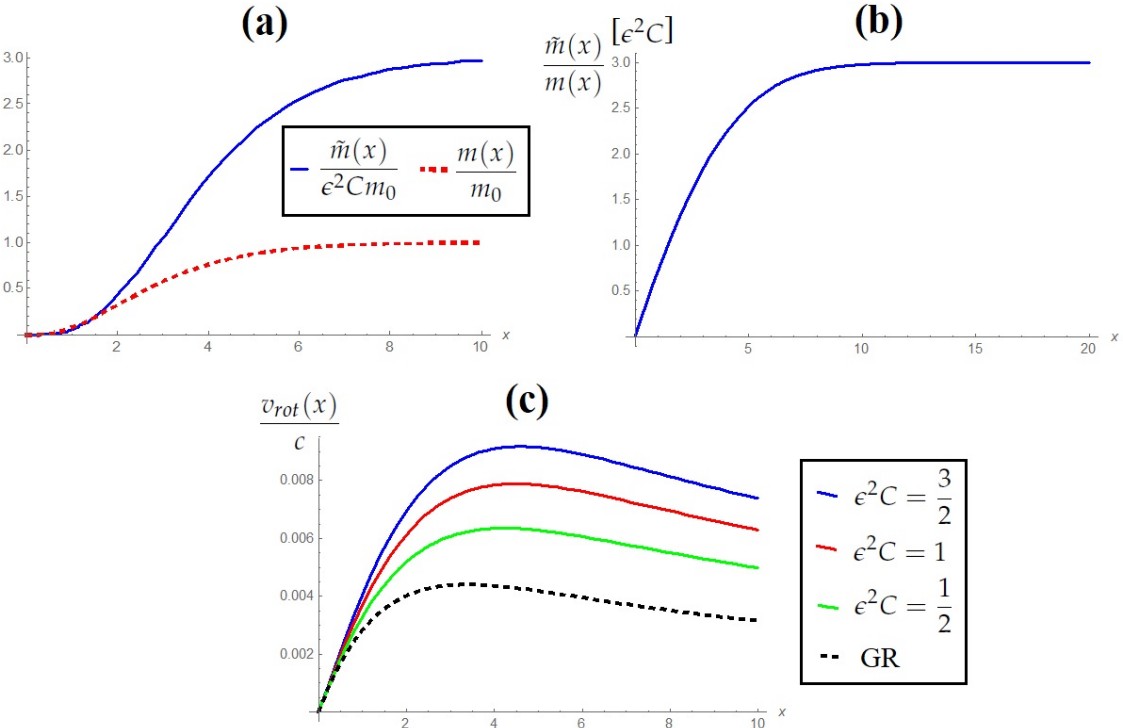

**Figure 2.** Exponential Profile Calculation. (**a**) ordinary and $\delta$ Matter vs. normalized radius, with $m_0 = 8\pi R^3 \rho_0$. From the center, more $\delta$ Matter is accumulated, but the behavior of both terms become similar when we distance from the center; (**b**) $\frac{\tilde{m}(x)}{m(x)}$ (in units of $\epsilon^2 C$) vs. normalized radius. We verify the last conclusion. In the center, the relation is almost linear, $\frac{\tilde{m}(x)}{m(x)} \sim x$, and, at the end of the galaxy, it is like a constant, $\frac{\tilde{m}(x)}{m(x)} \to 3\epsilon^2 C$; (**c**) rotation velocity vs. normalized radius for different values of $\epsilon^2 C$. The Black-Dashed line corresponds to GR case, so the $\epsilon^2 C$ value indicates the contribution of $\delta$ Matter. The similar behavior of both elements and the additional mass from $\delta$ Matter produce an amplifying effect in the rotation velocity. In these calculations, we have used $\kappa \rho_0 R^2 = 10^{-4}$.

### 4.3.3. Einasto Profile

The Einasto profile is a spherically symmetric distribution used to describe many types of real system, like galaxies and DM halos (see for instance [31–33]). It is represented by a logarithmic power-law:

$$\frac{d\ln(\rho(r))}{d\ln r} \propto -r^\alpha,$$

with $\alpha > 0$, then $\rho(x) = \rho_0 e^{-x^\alpha}$. Thus, it is a most general case of the exponential profile and many simulations of galaxies have been done using this profile, where they obtained values of $\alpha$ given by $0.1 \le \alpha \le 1$ [33]. Evaluating in Equations (110) and (111), we have:

$$\frac{1}{x^2}\frac{\partial}{\partial x}\left(x^2\left(\frac{\partial\phi(x)}{\partial x}\right)\right) = \frac{\kappa R^2 \rho_0 e^{-x^\alpha}}{2}, \tag{121}$$

$$\frac{1}{x^2}\frac{\partial}{\partial x}\left(x^2\left(\frac{\partial\tilde{\phi}(x)}{\partial x}\right)\right) = -\frac{\kappa R^2 \rho_0 \alpha x^{\alpha-1} e^{-x^\alpha}}{2\left(\frac{\partial\phi}{\partial x}(x)\right)}\left(\tilde{\phi}(x) - C\left(1 + 2\epsilon^2\phi(x)\right)\right). \tag{122}$$

As in the exponential profile, we can solve Equations (121) and (122) to find $m(x)$ and $\tilde{m}(x)$[3]. Thus, using expressions (112) and (113), we can see that the appropriate initial conditions are given by $m(x) \approx \frac{4\pi R^3 \rho_0}{3} x^3$ and $\tilde{m}(x) \approx \frac{4\pi R^3 \rho_0 \epsilon^2 C \alpha}{(3+\alpha)} x^{\alpha+3}$. Clearly, we can verify that this result is reduced to the exponential case with $\alpha = 1$ and we need $\epsilon^2 C \sim O(1)$ to obtain enough $\delta$ Matter too. $m(x)$ and $\tilde{m}(x)$ are represented in Figure 3a for $\alpha = \{0.7, 0.4, 0.1\}$, and we represent the relation between $m(x)$ and $\tilde{m}(x)$ in Figure 3b. As in the exponential case, more $\delta$ Matter is accumulated far from the center. Actually, we have that $\tilde{m}(x)$ increases faster than $m(x)$, especially when $\alpha$ is smaller. However, far from the center, $\frac{\tilde{m}(x)}{m(x)} \to constant \leq 3\epsilon^2 C$ when $\alpha$ is close to 1. For smaller $\alpha$'s, it is more like a logarithmic behavior. Finally, in Figure 3c, we present the rotation velocity for $\epsilon^2 C = \left\{ \frac{1}{2}, 1, \frac{3}{2} \right\}$ and GR. Just like we expected, the rotation velocity is amplified by the $\delta$ Matter effect, in such a way that if $C$ is bigger, we have higher velocities.

Our conclusions in this case are similar to the exponential profile. The ordinary matter leads over $\delta$ Matter in the center, but rapidly the second one increase until it is completely dominant. Thus, $\delta$ Matter is also concentrated outside of the galactic nucleus in this case, but additionally we obtained a logarithmic contribution from $\delta$ Matter, producing an additional DM behavior to small values of $\alpha$. We note this in Figure 3c for $\alpha = 0.1$.

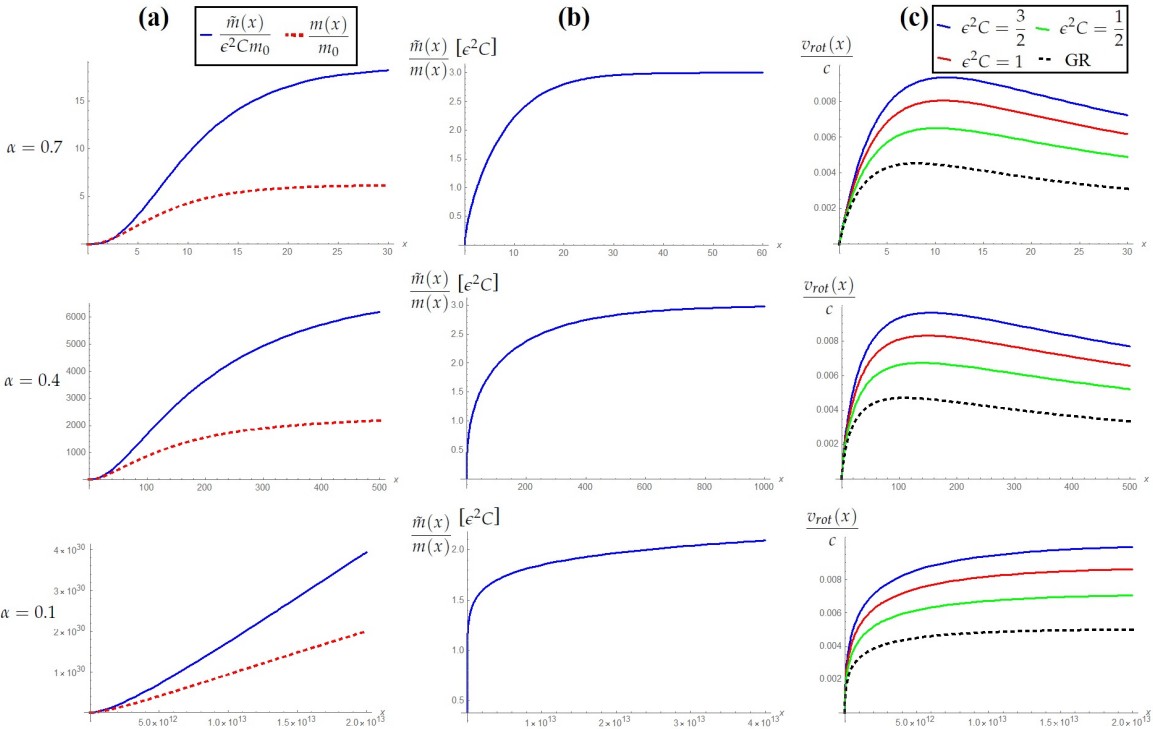

**Figure 3.** Einasto Profile Calculation for $\alpha = \{0.7, 0.4, 0.1\}$. (**a**) ordinary and $\delta$ Matter vs. normalized radius, with $m_0 = 8\pi R^3 \rho_0$. From the center, more $\delta$ Matter is accumulated, just like the exponential case; (**b**) $\frac{\tilde{m}(x)}{m(x)}$ (in units of $\epsilon^2 C$) vs. normalized radius. In fact, we have that $\tilde{m}(x)$ increases faster than $m(x)$ and it is faster when $\alpha$ is smaller. On the other side, at the end of the galaxy, it is like a constant too, with $\frac{\tilde{m}(x)}{m(x)} \to 3\epsilon^2 C$, but, for smaller $\alpha$, the behavior is more similar to a logarithmic function; (**c**) rotation velocity vs. normalized radius for different values of $\epsilon^2 C$. The Black-Dashed line corresponds to GR case, so the $\epsilon^2 C$ value indicates the contribution of $\delta$ Matter. In this case, we have an amplifying effect in the rotation velocity too, such that, if $\epsilon^2 C$ is bigger, we have higher velocities. In these calculations, we have used $\kappa \rho_0 R^2 = 4 \left(\frac{\alpha}{2}\right)^{\frac{2}{\alpha}} e^{\frac{2(1-\alpha)}{\alpha}} \times 10^{-4}$.

---

3   The only constant that we can not fix is $\phi_0 = \phi(0)$. Fortunately, this constant is irrelevant to find $m(x)$. This is true for an NFW profile too.

### 4.3.4. Navarro–Frenk–White Profile

The Navarro–Frenk–White (NFW) profile is another kind of distribution of the of DM halo (see, for instance, [34–36]), given by $\rho(x) = \frac{\rho_0}{x^\gamma (x+1)^{3-\gamma}}$. In pure DM simulations, $\gamma = 1$ is usually used; however, baryonic matter effects are expected, producing $1 \leq \gamma \leq 1.4$ [36]. Now, in this case, Equations (110) and (111) are:

$$\frac{1}{x^2} \frac{\partial}{\partial x} \left( x^2 \left( \frac{\partial \phi(x)}{\partial x} \right) \right) = \frac{\kappa R^2 \rho_0}{2 x^\gamma (x+1)^{3-\gamma}}, \tag{123}$$

$$\frac{1}{x^2} \frac{\partial}{\partial x} \left( x^2 \left( \frac{\partial \tilde{\phi}(x)}{\partial x} \right) \right) = -\frac{\kappa R^2 \rho_0 (\gamma + 3x)}{2 x^{\gamma+1} (x+1)^{4-\gamma} \left( \frac{\partial \phi}{\partial x}(x) \right)} \left( \tilde{\phi}(x) - C \left( 1 + 2 \epsilon^2 \phi(x) \right) \right). \tag{124}$$

We can also use Equations (112) and (113) to obtain the appropriate initial conditions. They are $m(x) \approx \frac{4\pi R^3 \rho_0}{3-\gamma} x^{3-\gamma}$ and $\tilde{m}(x) \approx \frac{4\pi \epsilon^2 C R^3 \rho_0 \gamma}{3-\gamma} x^{3-\gamma}$, then $\epsilon^2 C \sim O(1)$ too. After solving (123) and (124), we obtain $m(x)$ and $\tilde{m}(x)$, represented in Figure 4a, and the relation between $m(x)$ and $\tilde{m}(x)$ in Figure 4b for $\gamma = \{1, 1.2, 1.4\}$. Just like the exponential and Einasto cases, more $\delta$ Matter is accumulated far from the center. Compared with the other profiles, we have that $\tilde{m}(x)$ increases faster than $m(x)$ too, but in this case the relation between both kinds of matter is practically logarithmic, expected in a relation DM/Baryonic Matter. We know that $\gamma = 1$ correspond to just-dark-matter distribution and the other cases with $\gamma > 1$ consider a baryonic matter effect [36], which means that $\delta$ Gravity could give us a greater value of $\gamma$ than GR in a data simulation, so we obtain less ordinary Dark-Matter. In any case, we obtain the same conclusion; the ordinary matter leads over $\delta$ Matter in the center, but rapidly the second one increases until it is completely dominant. Thus, $\delta$ Matter is concentrated outside of the galactic nucleus.

The rotation velocity is presented in Figure 4c for $\epsilon^2 C = \left\{ \frac{1}{2}, 1, \frac{3}{2} \right\}$ and GR. In all our profiles, the rotation velocity is amplified by the $\delta$ Matter effect, such that, if $C$ is bigger, we have higher velocities.

From exponential, Einasto and NFW profiles, we saw that $\delta$ Matter produces an amplified effect to the ordinary matter, affecting the rotation velocity. Unfortunately, the $\delta$ Matter is principally concentrated outside of the galactic nucleus, as opposed to expected DM distribution. On the other side, from Einasto and NFW profiles, we observed a logarithmic relation between $m(x)$ and $\tilde{m}(x)$. That is:

$$\tilde{m}(x) \sim m(x) \ln(x). \tag{125}$$

In this way, we obtained an additional DM effect. Thus, we can divide the ordinary matter in Baryonic and DM; then, we have $\delta$ Baryonic and $\delta$ DM. In a cosmological level, the Einstein equations just take into account the ordinary matter, and $\delta$ Matter appears in the equation of $\tilde{g}_{\mu\nu}$ (see Equations (21) and (22)). All these mean that the quantity of ordinary (*real*) DM is less than GR. The rest of the DM effect is due to $\delta$ Matter components and the contribution is bigger when we get away from the center.

Now, if we compare the spherically homogeneous profile with the other ones, we note that the $\delta$ Matter effect is only produced by the distribution of the ordinary matter; the scale is not so important. Thus, if the DM is principally explained by $\delta$ DM, then this effect is only important when the distribution of ordinary matter is strongly non homogeneous. For example, we could find some Globular Clusters where the DM, $\delta$ Matter in this context, is less than Baryon Matter. Evidence of that has been found in [55,56], where an enormous quantity of DM is not necessary in the formation of some GCs. This computation will be developed in a future work.

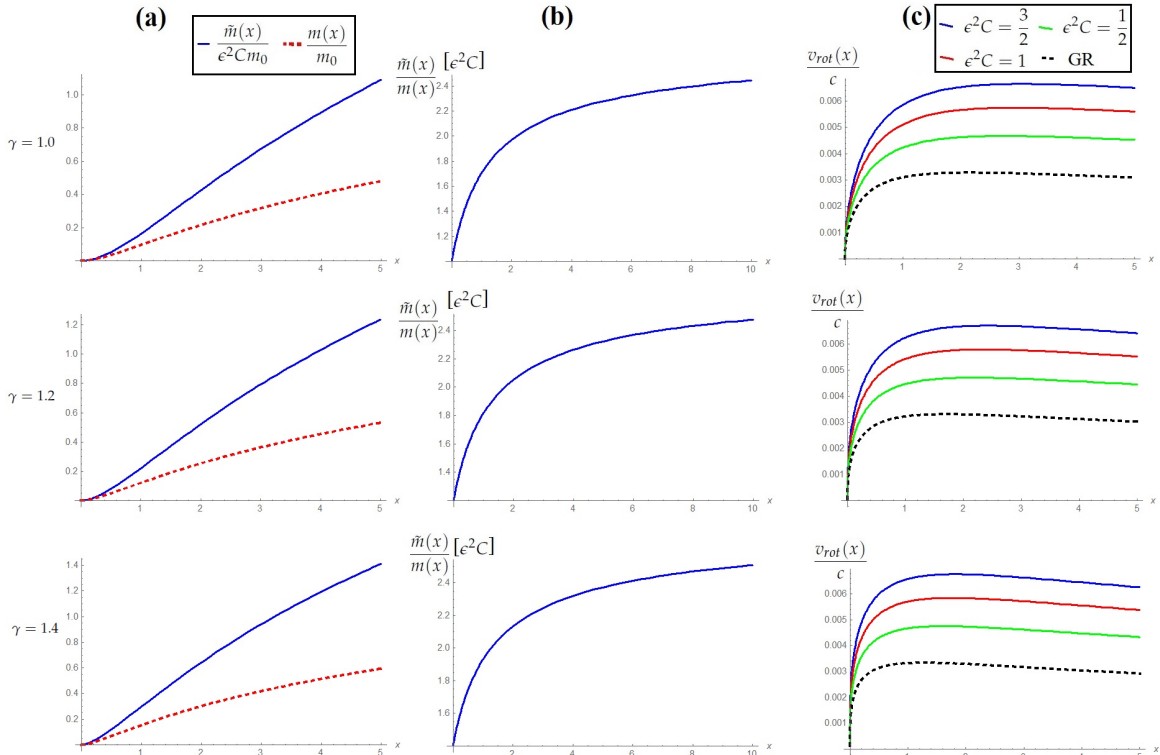

**Figure 4.** Navarro–Frenk–White (NFW) Profile Calculation for $\gamma = \{1, 1.2, 1.4\}$. (**a**) ordinary and $\delta$ Matter vs. normalized radius, with $m_0 = 8\pi R^3 \rho_0$. We have that more $\delta$ Matter is accumulated from the center, just like the exponential and Einasto cases; (**b**) $\frac{\tilde{m}(x)}{m(x)}$ (in units of $\epsilon^2 C$) vs. normalized radius. Compared with the other profiles, we have that $\tilde{m}(x)$ increases faster than $m(x)$, but, in this case, the relation between both masses is practically logarithmic; (**c**) rotation velocity vs. normalized radius for different values of $\epsilon^2 C$. The Black-Dashed line corresponds to GR cases and the other $\epsilon^2 C$ values indicate the contribution of $\delta$ Matter. Here, we verify the result in the other profiles, $\delta$ Matter amplifies the rotation velocity, such that, if $\epsilon^2 C$ is bigger, we have higher velocities. In these calculations, we have used $\kappa \rho_0 R^2 = \frac{(3-\gamma)^{3-\gamma}}{4(2-\gamma)^{2-\gamma}} \times 10^{-4}$.

A more accurate calculation could be developed using a specific profile for DM, an Einasto Profile with a small $\alpha$ or NFW profile with $\gamma = 1$ for example, and a second profile for Baryonic Matter. With these considerations, we can isolate the DM effect from Baryonic contribution, including $\delta$ Matter components. In [32], a multi-component Einasto profile was used. These computations will be also developed in a future work.

## 5. Schwarzschild Case

Until now, the accelerated expansion of the universe was explained without a cosmological constant and $\delta$ Matter was studied to explain the DM phenomenon in the rotation velocity in the galaxies. In this section, we will develop $\delta$ Gravity in a Schwarzschild geometry to study the principal phenomena used to test GR, the deflection of light by gravitational lensing and the perihelion precession; then, we will introduce the effect of $\delta$ Gravity in a black hole. Thus, in this case:

$$g_{\mu\nu}dx^\mu dx^\nu = -A(r)c^2 dt^2 + B(r)dr^2 + r^2 \left( d\theta^2 + \sin^2(\theta)d\phi^2 \right), \tag{126}$$

$$\tilde{g}_{\mu\nu}dx^\mu dx^\nu = -\tilde{A}(r)c^2 dt^2 + \tilde{B}(r)dr^2 + \tilde{F}(r)r^2 \left( d\theta^2 + \sin^2(\theta)d\phi^2 \right). \tag{127}$$

Before to solve $g_{\mu\nu}$ and $\tilde{g}_{\mu\nu}$, using the equations presented in Section 2, we need to fix the gauge and the correct boundary conditions. To satisfy Equation (A26), we will use the coordinate transformation:

$$
\begin{aligned}
X_1 &= (r - \mu)\sin(\theta)\cos(\phi), \\
X_2 &= (r - \mu)\sin(\theta)\sin(\phi), \\
X_3 &= (r - \mu)\cos(\theta),
\end{aligned}
$$

where $\mu = GM$. In this coordinate system, we have:

$$
g_{\mu\nu}dx^\mu dx^\nu = -A(r)c^2 dt^2 + \left(\frac{r}{r-\mu}\right)^2 d\mathbf{X}^2 + \left(\frac{B(r)}{(r-\mu)^2} - \frac{r^2}{(r-\mu)^4}\right)(\mathbf{X}\cdot d\mathbf{X})^2, \tag{128}
$$

$$
\tilde{g}_{\mu\nu}dx^\mu dx^\nu = -\tilde{A}(r)c^2 dt^2 + \tilde{F}(r)\left(\frac{r}{r-\mu}\right)^2 d\mathbf{X}^2 + \left(\frac{\tilde{B}(r)}{(r-\mu)^2} - \frac{\tilde{F}(r)r^2}{(r-\mu)^4}\right)(\mathbf{X}\cdot d\mathbf{X})^2, \tag{129}
$$

where $r = \mu + \sqrt{X_1^2 + X_2^2 + X_3^2}$. Equation (A26) is automatically satisfied, but this system is not convenient to work, so we will impose (A27) and then we will return to the standard coordinate system, given by (126) and (127). The additional condition to complete the gauge will be presented below.

## 5.1. Schwarzschild Solution

The correct boundary conditions, which give us the correct Minkowski limit, are $g_{\mu\nu} \to \eta_{\mu\nu}$ and $\tilde{g}_{\mu\nu} \to 0$ for $r \to \infty$[4]. Now, we can solve the equations of motion for the Schwarzschild metric. To simplify the problem, we will solve the equations in empty space. This means the region where $\tilde{T}_{\mu\nu} = T_{\mu\nu} = 0$. The solutions of our equations of motion (21) and (22) are:

$$
A(r) = 1 - \frac{2\mu}{r}, \tag{130}
$$

$$
B(r) = \frac{1}{1 - \frac{2\mu}{r}}, \tag{131}
$$

$$
\tilde{B}(r) = \frac{r^2(r-2\mu)\tilde{A}'(r) - 2\mu r\tilde{A}(r) + r(r-2\mu)(r-\mu)\tilde{F}'(r) + r(r-2\mu)\tilde{F}(r)}{(r-2\mu)^2} \tag{132}
$$

and survive the equation:

$$
r\tilde{A}''(r) + 2\tilde{A}'(r) - \mu\tilde{F}''(r) = 0, \tag{133}
$$

where $' = \frac{d}{dr}$. Equations (130) and (131) is the well-known Schwarzschild solution to Einstein equations, where we imposed $A(\infty) = B(\infty) = 1$, to obtain $g_{\mu\nu} \to \eta_{\mu\nu}$, when $r \to \infty$. As we said previously, we need to fix the gauge for $\tilde{g}_{\mu\nu}$ to obtain an additional equation of $\tilde{A}(r)$ and $\tilde{F}(r)$. This equation comes from (A27), so:

$$
r^2(r-2\mu)\tilde{A}''(r) + 4r(r-2\mu)\tilde{A}'(r) - 4\mu\tilde{A}(r) + r(r-2\mu)(r-\mu)\tilde{F}''(r) + 4(r-\mu)^2\tilde{F}'(r) = 0. \tag{134}
$$

---

[4]　In [19], we suggested that the boundary condition is given by $\tilde{g}_{\mu\nu} \to \eta_{\mu\nu}$ for $r \to \infty$ (see Equation (48) in the reference), but recently we noticed that the conditions presented in this paper are the correct choice. This is because $g_{\mu\nu} \to \eta_{\mu\nu}$, $\tilde{g}_{\mu\nu} \equiv \delta g_{\mu\nu}$ and $\delta\eta_{\mu\nu} = 0$, so it is natural to use $\tilde{g}_{\mu\nu} \to 0$ for $r \to \infty$.

Therefore, the general solution of (133) and (134) is given by:

$$\tilde{F}(r) \;=\; \tilde{F}_1 - \int_\infty^r \frac{u^3\,(u-2\mu)\,\tilde{A}''(u) + 2u\,(u+\mu)\,(u-2\mu)\,\tilde{A}'(u) - 4\mu\tilde{A}(u)}{4\mu\,(u-\mu)^2}\,du, \tag{135}$$

$$\tilde{A}(r) \;=\; \frac{\tilde{A}_1\mu(r-\mu)}{r^2} + \frac{\tilde{A}_2(r^2-2\mu^2)}{r^2} + \tilde{A}_3\mu\frac{2\mu + (r-\mu)\ln\left(1-\frac{2\mu}{r}\right)}{r^2}, \tag{136}$$

where $\tilde{A}_1$, $\tilde{A}_2$ and $\tilde{A}_3$ are integration constants. By the boundary conditions, we have to impose the conditions $\tilde{A}(\infty) = \tilde{B}(\infty) = \tilde{F}(\infty) = 0$ to obtain $\tilde{g}_{\mu\nu} \to 0$, when $r \to \infty$. These conditions just means that $\tilde{A}_2 = 0$ and $\tilde{F}_1 = 0$. Then, the solutions are ($\tilde{A}_1 = -2a_0$ and $\tilde{A}_3 = -a_1$):

$$\tilde{A}(r) \;=\; -\frac{2a_0\mu(r-\mu)}{r^2} - a_1\mu\frac{2\mu + (r-\mu)\ln\left(1-\frac{2\mu}{r}\right)}{r^2}, \tag{137}$$

$$\tilde{F}(r) \;=\; \frac{2a_0\mu}{r} - a_1\frac{2\mu + (r-\mu)\ln\left(1-\frac{2\mu}{r}\right)}{r}, \tag{138}$$

$$\tilde{B}(r) \;=\; \frac{2a_0\mu(r-\mu)}{(r-2\mu)^2} - a_1\frac{2\mu(r-2\mu) + (r^2-3\mu r+\mu^2)\ln\left(1-\frac{2\mu}{r}\right)}{(r-2\mu)^2}. \tag{139}$$

Thus, we have three parameters: $\mu$ comes from the ordinary metric components and represents the mass of a massive object (planets, stars, black holes, etc.). Finally, we have $a_0$ and $a_1$. These parameters are adimensional and represent the correction by $\delta$ Gravity. Later, we will understand the physical meaning of these parameters.

Now, we must remember that Equations (130), (131) and (137)–(139) correspond to the solution in the region without matter. This means that $r > R$, where $R$ is the radius of a star for example. Generally, the Newtonian approximation can be used, so that $R \gg 2\mu$. However, the logarithmic solution could be important in black holes, where the Newtonian approximation is not valid. Thus, considering the leading order in $\frac{\mu}{r}$, the solution is reduced to:

$$A(r) \;=\; 1 - \frac{2\mu}{r}, \tag{140}$$

$$B(r) \;=\; 1 + \frac{2\mu}{r} + O\left(\left(\frac{2\mu}{r}\right)^2\right), \tag{141}$$

$$\tilde{A}(r) \;=\; -\frac{2a_0\mu}{r} + O\left(\left(\frac{2\mu}{r}\right)^2\right), \tag{142}$$

$$\tilde{F}(r) \;=\; \frac{2a_0\mu}{r} + O\left(\left(\frac{2\mu}{r}\right)^2\right), \tag{143}$$

$$\tilde{B}(r) \;=\; \frac{2a_0\mu}{r} + O\left(\left(\frac{2\mu}{r}\right)^2\right). \tag{144}$$

Notice that $a_1$ disappears in Equations (140)–(144). This means that this parameter is only important in a Post-Newtonian approximation. We will use these expressions in the next section to describe the Gravitational Lensing effect.

*5.2. Gravitational Lensing*

To describe this phenomenon, we need the null geodesic, in our case, given by (36). Then, we will consider a coordinate system where $\theta = \frac{\pi}{2}$ such that the trajectory is given by Figure 5

(see, for instance, [54]). The geodesic equations are complicated, but, with some work, we may reduce it to:

$$\frac{dt}{du} = \frac{E}{A(r) + \tilde{A}(r)}, \tag{145}$$

$$\left(\frac{dr}{du}\right)^2 = \frac{E^2}{(A(r) + \tilde{A}(r))(B(r) + \tilde{B}(r))} - \frac{J^2}{r^2(B(r) + \tilde{B}(r))(1 + \tilde{F}(r))}, \tag{146}$$

$$\frac{d\phi}{du} = \frac{J}{r^2(1 + \tilde{F}(r))}, \tag{147}$$

where $u$ is the trajectory parameter such that $x^\mu = x^\mu(u)$ and $E$ and $J$ are constants of motion. From Equation (145), we can see that $t \to Eu$ when $r \to \infty$. On the other side, Equation (146) defines a *effective potential* given by:

$$V_{eff}(r) = 1 - \frac{1}{(A(r) + \tilde{A}(r))(B(r) + \tilde{B}(r))} + \frac{(J/E)^2}{r^2(B(r) + \tilde{B}(r))(1 + \tilde{F}(r))}, \tag{148}$$

such that:

$$v_r^2(r) \equiv \frac{1}{E^2}\left(\frac{dr}{du}\right)^2 = 1 - V_{eff}(r). \tag{149}$$

Using Equations (130)–(132) and (137)–(139), we can see that $V_{eff}(\infty) = 0$ and $v_r(\infty) = 1$. To obtain a plot of $V_{eff}(r)$, we need to fix $a_0$ and $a_1$. From Figure 5, we know that $\phi(r)$ is necessary to study the gravitational lensing. For this, we use Equations (146) and (147) to obtain:

$$\left(r^2\left(\frac{d\phi}{dr}(r)\right)\right)^{-2} = \frac{1 + \tilde{F}(r)}{B(r) + \tilde{B}(r)}\left(\frac{E^2(1 + \tilde{F}(r))}{J^2(A(r) + \tilde{A}(r))} - \frac{1}{r^2}\right)$$

$$\left(y^2\left(\frac{d\phi}{dy}(y)\right)\right)^{-2} = \frac{1 + \tilde{F}(r_0 y)}{B(r_0 y) + \tilde{B}(r_0 y)}\left(\frac{(1 + \tilde{F}(r_0 y))(A(r_0) + \tilde{A}(r_0))}{(1 + \tilde{F}(r_0))(A(r_0 y) + \tilde{A}(r_0 y))} - \frac{1}{y^2}\right), \tag{150}$$

where $y = \frac{r}{r_0} \geq 1$ is a normalized radius with $r_0$ the minimal radius, given by $\frac{dr}{du}|_{r=r_0} = 0$. Thus:

$$\left(\frac{J}{E}\right)^2 = r_0^2\frac{1 + \tilde{F}(r_0)}{A(r_0) + \tilde{A}(r_0)}.$$

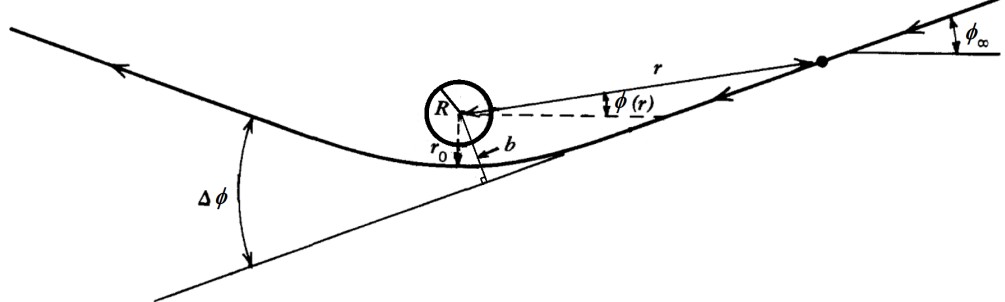

**Figure 5.** Trajectory by gravitational lensing. $R$ is the radius of the star, $r_0$ is the minimal distance to the star, $b$ is the impact parameter, $\phi_\infty$ is the incident direction and $\Delta\phi$ is the deflection of light.

Then, the deflection of light can be obtained solving (150). However, the approximation $r_0 >> 2\mu$ is usually used to obtain an explicit result of $\Delta\phi$ when the matter source is not dense enough. Thus, with this approximation, we obtain:

$$\left(\frac{d\phi}{dy}(y)\right)^{-2} \simeq y^2\left(y^2-1\right)\left(1-\frac{\epsilon}{y}-\frac{(1+2a_0)\,y\epsilon}{1+y}\right) + a_0 O\left(\epsilon^2\right), \tag{151}$$

where $\epsilon = \frac{2\mu}{r_0}$. We notice that this expression to first order on $\epsilon$ is exact in GR, but in $\delta$ Gravity we have higher order terms. Now, to obtain the deflection of light, we develop Equation (151) such that:

$$\frac{d\phi}{dy}(y) \simeq \pm\frac{1}{y\sqrt{y^2-1}}\left(1+\frac{\epsilon}{2y}+\frac{(1+2a_0)\,y\epsilon}{2\,(1+y)}\right)+O\left(\epsilon^2\right),$$

$$\phi(y)-\phi_\infty \simeq \pm\int_y^\infty\frac{dy'}{y'\sqrt{y'^2-1}}\left(1+\frac{\epsilon}{2y'}+\frac{(1+2a_0)\,y'\epsilon}{2\,(1+y')}\right)+O\left(\epsilon^2\right). \tag{152}$$

We want to describe a complete trajectory, so the photon start from $\phi(y=\infty)$ up to $\phi(y=1)$ and then back again to $\phi(y=\infty)$ (see Figure 5). In addition, if the trajectory were a straight line, this would equal just $\pi$. All of this means that the deflection of light is:

$$\begin{aligned}
\Delta\phi &= 2|\phi(1)-\phi_\infty|-\pi \\
&\simeq 2\left|\int_1^\infty\frac{dy'}{y'\sqrt{y'^2-1}}\left(1+\frac{\epsilon}{2y'}+\frac{(1+2a_0)\,y'\epsilon}{2\,(1+y')}\right)\right|-\pi+O\left(\epsilon^2\right) \\
&\simeq 2\,(1+a_0)\,\epsilon+O\left(\epsilon^2\right) \\
&\simeq \frac{4\mu\,(1+a_0)}{r_0}+O\left(\left(\frac{2\mu}{r_0}\right)^2\right).
\end{aligned} \tag{153}$$

From GR, $\Delta\phi = \frac{4\mu}{r_0}$. Thus, in our modified gravity, we have an additional term given by $\frac{4\mu a_0}{r_0}$. On the other side, we have an experimental value $\Delta\phi_{Exp} = 1.761'' \pm 0.016''$ for the sun [37] and it is very close to the prediction of GR. This means that, to satisfy the experimental value with $\delta$ Gravity, it is necessary that our additional term provide a very small correction, such that:

$$\begin{aligned}
\left|\frac{4\mu a_0}{r_0}\right| &= 1.761''|a_0| < 0.016'', \\
|a_0| &< 0.009086. \tag{154}
\end{aligned}$$

From Equation (153), we can see that $a_0$ represents an additional mass by $\delta$ Matter given by $M_{add} = a_0 M$, where $M$ is the solar mass. Thus, the result of (154) tells us that $\delta$ Matter must be less than 1% close to the Sun. In [57], it was estimated observationally that the DM mass in the sphere within Saturn's orbit should be less than $1.7 \times 10^{-10}\%$. On the other side, we expect that $\delta$ Matter will be more important in a galactic scale and explain a part of DM.

### 5.3. Perihelion Precession

In the last section, we used the null geodesic to compute the deflection of light. However, if we want to study the trajectory of a massive object, we need equations in (30). These equations, for $\theta = \frac{\pi}{2}$, are given by (see for instance [54]):

$$\left(\frac{dr}{du}\right)^2 = \frac{1}{B(r)}\left(\left(A(r) - \frac{J^2}{r^2\left(\frac{E}{A(r)} + \left(\tilde{F}(r) - \frac{\tilde{A}(r)}{A(r)}\right)\left(\frac{dt}{du}\right)\right)^2}\right)\left(\frac{dt}{du}\right)^2 - 1\right), \quad (155)$$

$$\frac{d\phi}{du} = \frac{J\left(\frac{dt}{du}\right)}{r^2\left(\frac{E}{A(r)} + \left(\tilde{F}(r) - \frac{\tilde{A}(r)}{A(r)}\right)\left(\frac{dt}{du}\right)\right)}, \quad (156)$$

where $\frac{dt}{du}$ obey a fifth order equation:

$$\left(1 + \frac{1}{2}\left(\frac{2\tilde{A}(r)}{A(r)} - \frac{\tilde{B}(r)}{B(r)} + \left(A(r)\left(\frac{\tilde{B}(r)}{B(r)} - \frac{\tilde{A}(r)}{A(r)}\right)\right.\right.\right.$$
$$\left.\left.\left. + \frac{J^2\left(\tilde{F}(r) - \frac{\tilde{B}(r)}{B(r)}\right)}{r^2\left(\frac{E}{A(r)} + \left(\tilde{F}(r) - \frac{\tilde{A}(r)}{A(r)}\right)\left(\frac{dt}{du}\right)\right)^2}\right)\left(\frac{dt}{du}\right)^2\right)\right)\left(\frac{dt}{du}\right) = \frac{E}{A(r)}. \quad (157)$$

These equations are very difficult to solve, but in the approximation $r_0 \gg 2\mu$ with $r_0$ the minimal radius, we have:

$$\frac{1}{E}\left(\frac{dt}{du}\right) \simeq 1 + \left(1 - a_0\left(j^2 - \frac{1}{2}\right)\right)\frac{\epsilon}{y}$$
$$+ \left(1 + \frac{a_0}{4y^2}\left(j^2\left(4y^3 - 10y^2 + 1\right) + y^2\left(4y - 1\right)\right) + \frac{a_0^2}{4}\left(2j^2 + 1\right)\left(4y + 6j^2 - 3\right)\right)\frac{\epsilon^2}{y^2} + O\left(\epsilon^3\right), \quad (158)$$

$$\left(\frac{dr}{du}\right)^2 \simeq j^2\left(1 - \frac{1}{y^2}\right)\left[1 - \frac{1 + a_0\left(2j^2 + 3\right)}{y}\epsilon - \frac{y\left(1 + j^2 + a_0\left(1 + 2j^2\right)\right)}{j^2\left(1 + y\right)}\epsilon\right.$$
$$+ \frac{a_0}{4}\left(\frac{2j^2}{y^4} - \frac{4j^2 - a_0\left(28j^4 + 44j^2 + 19\right)}{y^2} + \frac{8\left(1 + j^2 + a_0\left(2j^2 + 1\right)\right)}{y}\right.$$
$$\left.\left. + \frac{2 + 6j^2 + a_0\left(1 + 12j^2 + 4j^4\right)}{j^2} + \frac{4\left(1 + 3j^2 + 2j^4 + a_0\left(1 + 2j^2\right)^2\right)}{j^2\left(1 + y\right)}\right)\epsilon^2 + O\left(\epsilon^3\right)\right], \quad (159)$$

$$\frac{d\phi}{du} \simeq \frac{j}{r_0 y^2}\left[1 - a_0\left(j^2 + \frac{3}{2}\right)\frac{\epsilon}{y}\right.$$
$$\left. - a_0\left(\frac{5}{4} - y - \frac{j^2\left(2y - 1\right)\left(2y^2 - 2y - 1\right)}{4y^2} - a_0\left(3j^4 + 2j^2\left(2 + y\right) + y + \frac{5}{4}\right)\right)\frac{\epsilon^2}{y^2} + O\left(\epsilon^3\right)\right], \quad (160)$$

where $y = \frac{r}{r_0}$, $\epsilon = \frac{2\mu}{r_0}$, $j = \frac{J}{r_0}$ and $\frac{dr}{du}|_{r=r_0} = 0$, so:

$$E^2 \simeq 1 + j^2 - \left(1 + j^2 + a_0\left(2j^2 + 1\right)\right)\epsilon + \frac{a_0}{2}\left(1 + 3j^2 + a_0\left(2j^4 + 6j^2 + \frac{1}{2}\right)\right)\epsilon^2 + O(\epsilon^3).$$

From Equations (159) and (160), we obtain:

$$\left(\frac{d\phi}{dy}(y)\right)^2 \simeq y^{-2}\left(y^2 - 1\right)^{-1}\left[1 + \left(1 + \frac{y^2\left(1 + j^2 + a_0\left(2j^2 + 1\right)\right)}{j^2\left(1 + y\right)}\right)\frac{\epsilon}{y}\right. \quad (161)$$
$$+ \left(\frac{1 + \frac{a_0}{2}}{y^2} + \frac{\left(1 + j^2 + a_0\left(2j^2 + 1\right)\right)^2}{j^4\left(y + 1\right)^2} - \frac{2\left(1 + a_0\left(j^2 + 1\right)\right)\left(1 + j^2 + a_0\left(2j^2 + 1\right)\right)}{j^4\left(y + 1\right)}\right.$$
$$\left.\left. + \frac{4\left(j^2 + 1\right)^2 + 2a_0\left(5j^4 + 11j^2 + 4\right) - a_0^2\left(4j^6 - 4j^4 - 15j^2 - 4\right)}{4j^4}\right)\epsilon^2 + O\left(\epsilon^3\right)\right].$$

However, it is useful to rewrite it using:

$$\lambda = \frac{\lambda_0}{y}, \text{ with } \lambda_0 \simeq \frac{2j^2}{\epsilon\left(1 + a_0\left(2j^2 + 1\right) - 2a_0\left(1 + j^2 + a_0\left(2j^2 + 1\right)\right)\epsilon\right)}.$$

That is:

$$\left(\frac{d\lambda}{d\phi}(\phi)\right)^2 \simeq \bar{e} - 1 + 2\lambda(\phi) - \left(1 - \frac{a_0\left(6 - a_0\left(4j^4 - 4j^2 - 7\right)\right)\varepsilon}{2\left(1 + a_0\left(2j^2 + 1\right)\right)}\right)\lambda^2(\phi) + \varepsilon\lambda^3(\phi)$$

$$+ \frac{a_0\varepsilon^2}{2}\lambda^4(\phi) + a_0 O\left(\varepsilon^3\right), \tag{162}$$

where:

$$\varepsilon \simeq \frac{\left[1 + a_0\left(2j^2 + 1\right) - 2a_0\left(1 + j^2 + a_0\left(2j^2 + 1\right)\right)\epsilon\right]\epsilon^2}{2j^2} \tag{163}$$

$$\bar{e} \simeq 1 + \frac{j^2\left[4j^2 - 4\left(1 + j^2 + a_0\left(2j^2 + 1\right)\right)\epsilon + a_0\left(2 + 6j^2 + a_0\left(4j^4 + 12j^2 + 1\right)\right)\epsilon^2\right]}{\left(1 + a_0\left(2j^2 + 1\right) - 2a_0\left(1 + j^2 + a_0\left(2j^2 + 1\right)\right)\epsilon\right)^2\epsilon^2}$$

$$\simeq 1 + \frac{2\left(E^2 - 1\right)}{\left(1 + a_0\left(2j^2 + 1\right) - 2a_0\left(1 + j^2 + a_0\left(2j^2 + 1\right)\right)\epsilon\right)\varepsilon}, \tag{164}$$

with $\bar{e}$ the orbital eccentricity. In GR, Equation (162) is exact to first order on $\varepsilon$ and the cubic term in $\lambda(\phi)$ explain the mercury's perihelion precession. In $\delta$ Gravity, we have high order corrections too, but they are practically suppressed when $r_0 >> 2\mu$. In addition, Equation (164) can be interpreted as an energy redefinition, such that the *new energy* is given by $\tilde{E}^2 \sim 1 + \frac{\left(E^2 - 1\right)}{\left(1 + a_0\left(2j^2 + 1\right)\right)}$, but this modification must be small to satisfy Equation (154), so the orbital movements are equal to GR. On the other side, other corrections appear when $r_0$ is smaller, close to $2\mu$. In that case, we must use the exact equations, given by Equations (155)–(157), but they are very complicated. However, we can try to solve them in a particular radius, for example $r_0$ where $\frac{dr}{du} = 0$. In that case, Equations (155) and (157) can be reduced to:

$$\left(A(r_0) - \frac{j^2}{\left(\frac{E}{A(r_0)} + \left(\tilde{F}(r_0) - \frac{\tilde{A}(r_0)}{A(r_0)}\right)\left(\frac{dt}{du}\right)\Big|_{r_0}\right)^2}\right)\left(\frac{dt}{du}\right)^2\Bigg|_{r_0} = 1, \tag{165}$$

$$\left(1 + \frac{\tilde{A}(r_0)}{A(r_0)} - \frac{\tilde{F}(r_0)}{2} + \frac{A(r_0)}{2}\left(\tilde{F}(r_0) - \frac{\tilde{A}(r_0)}{A(r_0)}\right)\left(\frac{dt}{du}\right)^2\Bigg|_{r_0}\right)\left(\frac{dt}{du}\right)\Bigg|_{r_0} = \frac{E}{A(r_0)}, \tag{166}$$

where $j = \frac{J}{r_0}$. They are a fourth and a third order equation in $\left(\frac{dt}{du}\right)\Big|_{r_0}$, respectively. Thus, by iteration, we can reduce the order of these equation, such as:

$$\left(\frac{dt}{du}\right)\Bigg|_{r_0} = \frac{E}{A(r_0)}\Sigma(r_0), \tag{167}$$

with:

$$\Sigma(r_0) = \frac{1 - f_1(r_0)f_2(r_0)}{\left(1 + \frac{\tilde{A}(r_0)}{A(r_0)} - \frac{\tilde{F}(r_0)}{2}\right)\left(1 - \frac{4}{3}f_1(r_0)f_2(r_0)\right) + f_1(r_0)}, \tag{168}$$

$$f_1(r_0) = \frac{3E^2\left(\frac{\tilde{A}(r_0)}{A(r_0)} - \tilde{F}(r_0)\right)}{2A(r_0)\left(\frac{E^2}{A(r_0)} - j^2 + \left(2 + \frac{\tilde{A}(r_0)}{A(r_0)}\right)\left(\frac{\tilde{A}(r_0)}{A(r_0)} - \tilde{F}(r_0)\right)\right)}, \tag{169}$$

$$f_2(r_0) = \frac{4\left(1 + \frac{\tilde{A}(r_0)}{A(r_0)} - \frac{\tilde{F}(r_0)}{2}\right)}{\frac{E^2}{A(r_0)} - j^2 + \left(2 + \frac{\tilde{A}(r_0)}{A(r_0)}\right)\left(\frac{\tilde{A}(r_0)}{A(r_0)} - \tilde{F}(r_0)\right)} \tag{170}$$

and the minimum radius can be obtained numerically from:

$$\left(\frac{E^2}{A(r_0)} - j^2 + \left(2 + \frac{\tilde{A}(r_0)}{A(r_0)}\right)\left(\frac{\tilde{A}(r_0)}{A(r_0)} - \tilde{F}(r_0)\right)\right)\Sigma^2(r_0)$$
$$-4\left(1 + \frac{\tilde{A}(r_0)}{A(r_0)} - \frac{\tilde{F}(r_0)}{2}\right)\Sigma(r_0) + 3 = 0, \tag{171}$$

where we have to use the exact solution of $A(r_0)$, $B(r_0)$, $\tilde{A}(r_0)$, $\tilde{B}(r_0)$ and $\tilde{F}(r_0)$ in Section 5.1. In the process, we imposed that Equations (165) and (166) possess one pole in common for $\Sigma(r_0)$. $r_0$ is an important element to understand the orbital trajectories, but we need to fix $a_0$ and $a_1$ to solve it. On the other side, we can verify that in the limit where $\left(\tilde{A}(r_0), \tilde{B}(r_0), \tilde{F}(r_0)\right) \to 0$, our results are reduced to GR, which is:

$$(168) \rightarrow \left.\left(\frac{dt}{du}\right)\right|_{r_0} = \frac{E}{A(r_0)} = \frac{E}{1 - \frac{2\mu}{r_0}},$$

$$(171) \rightarrow E^2 = A(r_0)\left(1 + j^2\right) \rightarrow r_0 = \frac{2\mu\left(1 + j^2\right)}{1 + j^2 - E^2}.$$

In conclusion, $\delta$ Gravity gives us important corrections to orbital trajectories, but they do not produce big differences with GR when the trajectory is far away from the Schwarzschild radius, $r_s = 2\mu$, on the condition that $a_0$ is small enough (See Equation (154)). This is always true for stars, planets and any low-density object. We saw that $a_0$ represents the $\delta$ Matter contribution; however, the physical meaning of $a_1$ is unknown yet. To solve this, the study of massive object is necessary. To finish, in the next section, we will introduce the analysis of Black Holes to complete the Schwarzschild case.

*5.4. Black Holes*

In Section 2.3, we saw that the proper time is defined using the metric $g_{\mu\nu}$, such as GR where $g_{\mu\nu}\dot{x}^\mu\dot{x}^\nu = -1$. Then, the proper time is given by (37). On the other side, the null geodesic of massless particles tells us that the space geometry is determined by both tensor fields, $g_{\mu\nu}$ and $\tilde{g}_{\mu\nu}$. This means that the three-dimensional metric in a Schwarzschild geometry is given by [58]:

$$\begin{aligned}
dl^2 &= \gamma_{ij}dx^i dx^j \\
&= \frac{A(r)\left(B(r) + \tilde{B}(r)\right)}{\left(A(r) + \tilde{A}(r)\right)}dr^2 + \frac{r^2 A(r)\left(1 + \tilde{F}(r)\right)}{\left(A(r) + \tilde{A}(r)\right)}\left(d\theta^2 + \sin^2(\theta)d\phi^2\right).
\end{aligned} \tag{172}$$

To guarantee that the three-dimensional metric is definite positive, we need:

$$\gamma_{11} > 0, \quad \begin{vmatrix} \gamma_{11} & \gamma_{12} \\ \gamma_{21} & \gamma_{22} \end{vmatrix} > 0 \text{ and } \begin{vmatrix} \gamma_{11} & \gamma_{12} & \gamma_{13} \\ \gamma_{21} & \gamma_{22} & \gamma_{23} \\ \gamma_{31} & \gamma_{32} & \gamma_{33} \end{vmatrix} > 0. \tag{173}$$

Thus, if we apply the conditions (173) plus $g_{00} < 0$ to Equation (172), we obtain:

$$A(r) > 0 \wedge f_A(r) > 0 \wedge f_B(r) > 0 \wedge f_F(r) > 0, \tag{174}$$

where:

$$
\begin{aligned}
f_A(r) &= A(r) + \tilde{A}(r) \\
&= 1 - \frac{2\mu}{r} - \frac{2a_0\mu\,(r - \mu)}{r^2} - a_1\mu\,\frac{2\mu + (r - \mu)\ln\left(1 - \frac{2\mu}{r}\right)}{r^2}, \tag{175}\\
f_B(r) &= B(r) + \tilde{B}(r) \\
&= \frac{r}{r - 2\mu} + \frac{2a_0\mu\,(r - \mu)}{(r - 2\mu)^2} - a_1\,\frac{2\mu\,(r - 2\mu) + (r^2 - 3\mu r + \mu^2)\ln\left(1 - \frac{2\mu}{r}\right)}{(r - 2\mu)^2}, \tag{176}\\
f_F(r) &= 1 + \tilde{F}(r) \\
&= 1 + \frac{2a_0\mu}{r} - a_1\,\frac{2\mu + (r - \mu)\ln\left(1 - \frac{2\mu}{r}\right)}{r}. \tag{177}
\end{aligned}
$$

We can see that these rules are automatically satisfied when $r \gg 2\mu$, so we must consider extreme cases to prove these conditions. In GR, they are reduced to $A(r) > 0$ and $B(r) > 0$, then $r > 2\mu$. This means that these rules define the black hole horizon, $r_H = 2\mu$. Therefore, Equation (174) defines a modified horizon to $\delta$ Gravity, but we need to fix $a_0$ and $a_1$ first.

Motivated by the result in Section 5.2, we know that $a_0$ is related to $\delta$ Matter, so $a_0 > 0$ and probably bigger than the result in (154) to consider a higher quantity of $\delta$ Matter (DM in this case). Just as an example, we can choose some combination:

- If $a_0 = 1$ and $a_1 = 1$, we have $r_H = 3.37\mu$,
- If $a_0 = 1$ and $a_1 = -1$, we have $r_H = 3.46\mu$.

In both cases, the horizon radius is given by $f_A(r_H) = 0$. However, if $|a_1| \gg 1$ and $a_0 \sim 1$, then the horizon is given by $f_B(r)$ ($a_1 > 0$) or $f_F(r)$ ($a_1 < 0$). In any case, we will obtain a horizon radius $\geq r_H = 2\mu$.

In GR, the event horizon radius is defined when $g_{rr}$ component of the metric is null. In fact, when we include Electric Charge and/or Angular Momentum in a black hole, inner and outer event horizons are produced and, additionally, an ergosphere appears, given by $g_{tt} = 0$, defining different regions around the black hole. In $\delta$ Gravity, we saw that the three-dimensional metric gives us the event horizon radiuses and, in the same way as GR, different regions are produced, but Electric Charge or Angular Momentum are not necessary. In a Schwarzschild black hole, these regions are produced whenever the conditions in (174) are violated. Now, the nature of these regions will depend of the value of $a_0$ and $a_1$.

In conclusion, we have that the effects of $\delta$ Gravity are represented by $a_0$ and $a_1$. $a_0$ has a clear physical meaning; it represents the quantity of $\delta$ Matter and could be considered as DM. On the other side, the meaning of $a_1$ is more difficult to define. This parameter only appears when we consider a highly massive object, as black holes, defining different kinds of regions around of this object. In that sense, it could redefine the concept of black holes. For that reason, black holes in $\delta$ Gravity will be studied in more detail in a future work.

## 6. Conclusions

We have proposed a modified model of gravity. It incorporates a new gravitational field $\tilde{g}_{\mu\nu}$ that transforms correctly under GCT and exhibits a new symmetry: the $\delta$ symmetry. The new action is invariant under these transformations. We call this new gravity model $\delta$ Gravity. In this paper, we studied $\delta$ Gravity at a classical level. For this, we were required to set up the following three issues.

First, we needed to find the equations of motion for $\delta$ Gravity. One of them are Einstein's equations, which gives us $g_{\mu\nu}$, and additionally we have the equation of $\tilde{g}_{\mu\nu}$. Secondly, we presented the modified geometry for this model, incorporating the new field $\tilde{g}_{\mu\nu}$, where the massless particles, like a photon for example, move in a null geodesic of the effective metric $\mathbf{g}_{\mu\nu} = g_{\mu\nu} + \tilde{g}_{\mu\nu}$. Third, we needed to fix the gauge for $g_{\mu\nu}$ and $\tilde{g}_{\mu\nu}$. For this, we developed the extended harmonic gauge given by (A26) and (A27).

In this paper, we studied three particular phenomenons in a cosmological level. In the first place, we present the calculation developed in [25]. Unlike [23] and [24], we found the exact solution for FLRW geometry in $\delta$ Gravity and preserving $\delta$ Matter, assuming a universe with just non-relativistic matter and radiation. Then, we verified that $\delta$ Gravity does not require DE to explain the accelerated expansion of the universe because a new scale factor $\tilde{R}(t)$ is defined with the effective metric, given by (66). Then, we computed the age of the Universe and it is practically the same as in GR and Planck. On the other side, our model ends in a Big-Rip and we computed that the universe has lived less than half of its life. In addition, we computed the $\delta$ Matter in the present, where the $\delta$ non-relativistic matter is 23% of the ordinary non-relativistic matter. This result may imply that DM is in part $\delta$ Matter. In addition, a very small quantity of $\delta$ radiation has been found.

In second place, we studied the Non-Relativistic case. In the Newtonian limit, we obtained a similar expression as in GR, where we have an effective potential. This potential depends on $\rho^{(0)}$ and $\tilde{\rho}^{(0)}$, where the last one corresponds to $\delta$ Matter. We found a relation between $\rho^{(0)}$ and $\tilde{\rho}^{(0)}$ and we used it in different density profiles for a galaxy to explain DM. In the first place, we can see that, in a spherically homogenous density, the $\delta$ Matter effect is completely null. If we consider this profile like a first approximation to a planet or a star, we conclude that in these cases do not have $\delta$ Matter contributions, so it is equivalent to GR. On the other side, with other kinds of densities, where the matter distribution changes in the space, we obtain important modifications. Particularly, we used the exponential, Einasto and NFW profiles, where we noted that $\delta$ Matter produces an amplifying effect in the total mass and rotation velocity in a galaxy. Considering all these, we can say that the $\delta$ Matter effect is only important when the distribution of ordinary matter is strongly non homogeneous, so the scale is not so important. Therefore, considering $\delta$ Matter as DM, large structures with a small quantity of DM should be found [55,56]. In addition, we saw that $\delta$ Matter has a special behavior, more similar to DM compared with its equivalent ordinary component. We can see this in Figures 3 and 4, where a logarithmic relation between $\delta$ and Ordinary Matter is observed, so $\delta$ Matter contribution is bigger when we get away from the center. A more complete calculation can be developed if we use a multi-components profile to simulate data of some galaxies [32]. In this way, we can isolate the different contributions: Ordinary Baryonic Matter, Ordinary DM, $\delta$ Baryonic Matter and $\delta$ DM. An analogous result must be obtained in the CMB Power Spectrum. With all these, we concluded that the DM effect could be explained with a considerably less quantity of Ordinary DM, considering that the principal source of this effect is $\delta$ DM. This result would explain the problematic detection of DM; however, a field theory description of $\delta$ Gravity is necessary to understand the nature of $\delta$ Matter.

Finally, we analyzed the Schwarzschild case outside matter. We found an exact solution for the equations of motion and used the Newtonian approximation in these solutions to find the deflection of light by the Sun. To explain the experimental data, the correction must be small, such that $\delta$ Matter is <1% of the total mass at a solar system scale. Then, we study the perihelion precession. The exact solution is very complicated because a fifth order equation must be solved. However, even in the Newtonian approximation, we can see interesting, but really small, corrections. Basically, $\delta$ Gravity does not have important corrections to GR for low-density object like stars or planets. This means that we need to study high-density objects to observe important effects by $\delta$ Gravity. To have an idea of how the trajectory of a massive particle is affected by massive objects, we solved the equations in the minimum radius. For that reason, we presented an introduction to Black Holes, where some conditions to guarantee that the three-dimensional metric is definite positive are studied. In the same way than

GR, the three-dimensional metric for $\delta$ Gravity gives us the inner and outer event horizon radiuses, defining different regions like an ergosphere whenever the conditions in (174) are violated, even in a Schwarzschild black hole. We understood that $a_0$ gives us the quantity of $\delta$ Matter (maybe DM), but the meaning of $a_1$ is more difficult to define. In any case, it is only relevant to highly massive object, so it is important to define these regions. Black holes in $\delta$ Gravity must be studied in more detail.

We have shown that $\mathbf{g}_{\mu\nu} = g_{\mu\nu} + \tilde{g}_{\mu\nu}$ is the graviton, whereas $\tilde{g}_{\mu\nu}$ is a ghost [19,38–41]. In the non ghost-free models, like $\delta$ Theories [22], the Hamiltonian is not bounded from below. On the other side, phantom cosmological models, produced by a ghost component, are used to explain the accelerated expansion of the Universe [26–30], but this background solution becomes unstable by the ghost field. However, some mechanisms can be implemented to restrict the ghost component, avoid the instability and resurrect some of these theories [42–46]. Examples of this are developed in the *Lee and Wick finite electrodynamic theory*, non-Hermitian models with *PT symmetry* and others where some restrictions to the configuration space are introduced to make the ghost modes harmless. Added to the quantum corrections limited to one loop, it is not clear whether this problem will persist or not in a diffeomorphism-invariant model as $\delta$ Gravity.

At this moment, we have studied some phenomenon (Expansion of the Universe, DM, the Deflection of Light, etc.) and introduced a preliminary discussion of others (Black Holes and Inflation). Further tests of the model must include the computation of the CMB power spectrum, the evolution and formation of large-scale structure in the universe and a more detailed analysis of DM. These works are in progress now.

**Author Contributions:** The original idea of this model was created by J.A., being Section 2 a compilation of the initial works of this model. An important part of this work, including the calculation and the numerical analysis in Sections 3–5, was presented in the PhD thesis of P.G., with the supervision of J.A. These calculations were improved in the last year. Both authors contributed equally in the formal analysis of this work.

**Funding:** The work of P.G. has been partially financed by Beca Doctoral Conicyt $N^0$ 21080490, Fondecyt 1110378, Anillo ACT 1102, Anillo ACT 1122 and CONICYT Programa de Postdoctorado FONDECYT $N^o$ 3150398. The work of J.A. is partially supported by Fondecyt 1110378, Fondecyt 1150390, Anillo ACT 1102 and Anillo ACT 11016.

**Acknowledgments:** The authors want to thank Francisco Prada, Radoslaw Wojtak and Nelson Videla for useful remarks.

**Conflicts of Interest:** The authors declare no conflict of interest.

## Abbreviations

The following abbreviations are used in this manuscript:

| | |
|---|---|
| DM | Dark Matter |
| DE | Dark Energy |
| GR | General Relativity |
| DGT | $\delta$ Gauge Theories |
| GCT | General Coordinates Transformation |
| ExGCT | Extended General Coordinates Transformation |
| NFW | Navarro–Frenk–White |
| FLRW | Friedmann–Lemaître–Robertson–Walker |

## Appendix A. Perfect Fluid

To parameterize a perfect fluid, the usual action is [59]:

$$S_0 = \int d^4x \sqrt{-g} \left( \frac{R}{2\kappa} - \mathbf{r}(1 + \varepsilon(\mathbf{r})) - \lambda_1(u^a u_a + 1) - \lambda_2 D_\alpha(\mathbf{r}U^\alpha) \right), \tag{A1}$$

where $\mathbf{r}$ is the number of particles per unit volume in the mean frame of reference of these particles, $\varepsilon(\mathbf{r})$ is the internal energy density per unit mass of the fluid, $u_a$ is the speed of the fluid in the local frame and $\lambda_1$ and $\lambda_2$ are Lagrange multipliers that ensure the normalization of $u_a$ and conservation

of mass, respectively. Finally, we have that $U_\alpha = e_\alpha^a u_a$, where $e_\alpha^a$ is the Vierbein. From this action, we can see that the independent variables are $g_{\mu\nu}$, $\mathbf{r}$, $u_a$, $\lambda_1$ and $\lambda_2$, where $e_\alpha^a$ depends on $g_{\mu\nu}$. Thus, our modified action is:

$$S = \int d^4x \sqrt{-g} \left( \frac{R}{2\kappa} - \mathbf{r}(1 + \varepsilon(\mathbf{r})) - \lambda_1 (u^a u_a + 1) - \lambda_2 D_\alpha (\mathbf{r} U^\alpha) \right.$$

$$\left. - \frac{1}{2\kappa} \left( G^{\alpha\beta} - \kappa T^{\alpha\beta} \right) \tilde{g}_{\alpha\beta} + \tilde{L}_M \right), \tag{A2}$$

$$\tilde{L}_M = -\tilde{\mathbf{r}} (1 + \varepsilon(\mathbf{r}) + \mathbf{r}\varepsilon'(\mathbf{r})) - \tilde{\lambda}_1 (u^a u_a + 1) - 2\lambda_1 u^a \tilde{u}_a - \tilde{\lambda}_2 D_\alpha (\mathbf{r} U^\alpha)$$

$$- \lambda_2 D_\alpha (\tilde{\mathbf{r}} U^\alpha + \mathbf{r} U_T^\alpha), \tag{A3}$$

with $\tilde{\mathbf{r}} = \tilde{\delta}\mathbf{r}$, $\varepsilon'(\mathbf{r}) = \frac{\partial \varepsilon}{\partial \mathbf{r}}(\mathbf{r})$, $\tilde{u}_a = \tilde{\delta} u_a$, $U_T^\alpha = e^{a\alpha} \tilde{u}_a$, $\tilde{\lambda}_1 = \tilde{\delta}\lambda_1$ and $\tilde{\lambda}_2 = \tilde{\delta}\lambda_2$ are new Lagrange multipliers. The energy-momentum tensor is:

$$T_{\mu\nu} = -\left(\mathbf{r}(1 + \varepsilon(\mathbf{r})) + \lambda_1(u^a u_a + 1) - \lambda_{2,\alpha}\mathbf{r} U^\alpha\right) g_{\mu\nu} - \frac{1}{2}\lambda_{2,\alpha}\mathbf{r} \left( \delta_\nu^\alpha U_\mu + \delta_\mu^\alpha U_\nu \right) \tag{A4}$$

and we have used $\frac{\delta e_\alpha^a}{\delta g_{\mu\nu}} = \frac{1}{4} \left( \delta_\alpha^\mu e^{a\nu} + \delta_\alpha^\nu e^{a\mu} \right)$. In addition, we can compute that:

$$\tilde{T}_{\mu\nu} = -\frac{1}{4}\lambda_{2,\beta}\mathbf{r} \left( \delta_\nu^\beta U^\alpha \tilde{g}_{\mu\alpha} + \delta_\mu^\beta U^\alpha \tilde{g}_{\nu\alpha} + 2g_{\mu\nu} U^\alpha \tilde{g}_\alpha^\beta \right) \tag{A5}$$

$$- \left( \mathbf{r}(1 + \varepsilon(\mathbf{r})) + \lambda_1(u^a u_a + 1) - \lambda_{2,\rho}\mathbf{r} U^\rho \right) \tilde{g}_{\mu\nu}$$

$$- \frac{1}{2}\tilde{\lambda}_{2,\alpha}\mathbf{r} \left( \delta_\nu^\alpha U_\mu + \delta_\mu^\alpha U_\nu \right) - \frac{1}{2}\lambda_{2,\alpha}\tilde{\mathbf{r}} \left( \delta_\nu^\alpha U_\mu + \delta_\mu^\alpha U_\nu \right) - \frac{1}{2}\lambda_{2,\alpha}\mathbf{r} \left( \delta_\nu^\alpha U_\mu^T + \delta_\mu^\alpha U_\nu^T \right)$$

$$- \left( \tilde{\mathbf{r}}(1 + \varepsilon(\mathbf{r}) + \mathbf{r}\varepsilon'(\mathbf{r})) + \tilde{\lambda}_1(u^a u_a + 1) + 2\lambda_1 u^a \tilde{u}_a - \tilde{\lambda}_{2,\alpha}\mathbf{r} U^\alpha - \lambda_{2,\alpha}(\tilde{\mathbf{r}} U^\alpha + \mathbf{r} U_T^\alpha) \right) g_{\mu\nu}.$$

Then, we have a modified action with ten independent variables: $g_{\mu\nu}$, $\mathbf{r}$, $u_a$, $\lambda_1$, $\lambda_2$, $\tilde{g}_{\mu\nu}$, $\tilde{\mathbf{r}}$, $\tilde{u}_a$, $\tilde{\lambda}_1$ and $\tilde{\lambda}_2$. Thus, we must solve Equations (21) and (22) using (A4) and (A5) to obtain $g_{\mu\nu}$ and $\tilde{g}_{\mu\nu}$. Fortunately, we can use the equations of motion for $\mathbf{r}$, $u_a$, $\lambda_1$, $\lambda_2$, $\tilde{\mathbf{r}}$, $\tilde{u}_a$, $\tilde{\lambda}_1$ and $\tilde{\lambda}_2$. These equations can be reduced to:

$$u^a u_a + 1 = 0, \tag{A6}$$

$$D_\alpha (\mathbf{r} U^\alpha) = 0, \tag{A7}$$

$$2\lambda_1 u^a - \mathbf{r} e^{a\alpha}\lambda_{2,\alpha} = 0, \tag{A8}$$

$$1 + \varepsilon(\mathbf{r}) + \mathbf{r}\varepsilon'(\mathbf{r}) - U^\alpha \lambda_{2,\alpha} = 0, \tag{A9}$$

$$u^a \tilde{u}_a = 0, \tag{A10}$$

$$D_\alpha \left( \tilde{\mathbf{r}} U^\alpha + \mathbf{r} U_T^\alpha - \frac{1}{2}\mathbf{r}\tilde{g}^{\alpha\beta}U_\beta + \frac{1}{2}\mathbf{r}\tilde{g}_\beta^\beta U^\alpha \right) = 0, \tag{A11}$$

$$2\tilde{\lambda}_1 u^a + 2\lambda_1 \tilde{u}^a - e^{a\alpha} \left( \mathbf{r}\tilde{\lambda}_{2,\alpha} + \tilde{\mathbf{r}}\lambda_{2,\alpha} - \frac{1}{2}\tilde{g}_\alpha^\beta \mathbf{r}\lambda_{2,\beta} \right) = 0, \tag{A12}$$

$$\tilde{\mathbf{r}} \left( 2\varepsilon'(\mathbf{r}) + \mathbf{r}\varepsilon''(\mathbf{r}) \right) - U^\alpha \tilde{\lambda}_{2,\alpha} - U_T^\alpha \lambda_{2,\alpha} + \frac{1}{2}U_\beta \tilde{g}^{\alpha\beta}\lambda_{2,\alpha} = 0. \tag{A13}$$

Now, we can use these equations to simplify the expressions in (A4) and (A5), eliminating the Lagrange multipliers rewriting Equations (A8), (A9), (A12) and (A13) as:

$$\lambda_1 = -\frac{1}{2}\mathbf{r} \left( 1 + \varepsilon(\mathbf{r}) + \mathbf{r}\varepsilon'(\mathbf{r}) \right), \tag{A14}$$

$$\lambda_{2,\mu} = -\left( 1 + \varepsilon(\mathbf{r}) + \mathbf{r}\varepsilon'(\mathbf{r}) \right) U_\mu, \tag{A15}$$

$$\tilde{\lambda}_1 = -\frac{1}{2}\tilde{\mathbf{r}} \left( 1 + \varepsilon(\mathbf{r}) + 3\mathbf{r}\varepsilon'(\mathbf{r}) + \mathbf{r}^2 \varepsilon''(\mathbf{r}) \right), \tag{A16}$$

$$\tilde{\lambda}_{2,\mu} = -\tilde{\mathbf{r}} \left( 2\varepsilon'(\mathbf{r}) + \mathbf{r}\varepsilon''(\mathbf{r}) \right) U_\mu - \left( 1 + \varepsilon(\mathbf{r}) + \mathbf{r}\varepsilon'(\mathbf{r}) \right) \left( U_\mu^T + \frac{1}{2}\tilde{g}_\mu^\beta U_\beta \right). \tag{A17}$$

Then, the equations that survive are:

$$U^\alpha U_\alpha + 1 = 0, \tag{A18}$$

$$D_\alpha(\mathbf{r} U^\alpha) = 0, \tag{A19}$$

$$(1 + \varepsilon(\mathbf{r}) + \mathbf{r}\varepsilon'(\mathbf{r})) \, U^\alpha D_\alpha U_\mu + \left(\delta_\mu^\alpha + U^\alpha U_\mu\right) (2\varepsilon'(\mathbf{r}) + \mathbf{r}\varepsilon''(\mathbf{r})) \, \partial_\alpha \mathbf{r} = 0, \tag{A20}$$

$$U^\alpha U_\alpha^T = 0, \tag{A21}$$

$$D_\alpha \left( \tilde{\mathbf{r}} U^\alpha + \mathbf{r} U_T^\alpha - \frac{1}{2}\mathbf{r}\tilde{g}^{\alpha\beta}U_\beta + \frac{1}{2}\mathbf{r}\tilde{g}_\beta^\beta U^\alpha \right) = 0, \tag{A22}$$

$$\tilde{\mathbf{r}} \left(2\varepsilon'(\mathbf{r}) + \mathbf{r}\varepsilon''(\mathbf{r})\right) U^\alpha D_\alpha U_\mu + (1 + \varepsilon(\mathbf{r}) + \mathbf{r}\varepsilon'(\mathbf{r})) \left( U_T^\alpha - \frac{1}{2}\tilde{g}^{\alpha\beta}U_\beta \right) D_\alpha U_\mu$$

$$+ (1 + \varepsilon(\mathbf{r}) + \mathbf{r}\varepsilon'(\mathbf{r})) \, U^\alpha D_\alpha \left( U_\mu^T + \frac{1}{2}\tilde{g}_{\mu\beta}U^\beta \right) + \frac{1}{2}(1 + \varepsilon(\mathbf{r}) + \mathbf{r}\varepsilon'(\mathbf{r})) \, U^\alpha U^\beta D_\mu \tilde{g}_{\alpha\beta},$$

$$+ \left( \left( U_T^\alpha - \frac{1}{2}\tilde{g}^{\alpha\beta}U_\beta \right) U_\mu + U^\alpha \left( U_\mu^T + \frac{1}{2}\tilde{g}_{\mu\beta}U^\beta \right) \right) (2\varepsilon'(\mathbf{r}) + \mathbf{r}\varepsilon''(\mathbf{r})) \, \partial_\alpha \mathbf{r}$$

$$+ \left( \delta_\mu^\alpha + U^\alpha U_\mu \right) \left( \tilde{\mathbf{r}} \left(3\varepsilon''(\mathbf{r}) + \mathbf{r}\varepsilon'''(\mathbf{r})\right) \partial_\alpha \mathbf{r} + (2\varepsilon'(\mathbf{r}) + \mathbf{r}\varepsilon''(\mathbf{r})) \partial_\alpha \tilde{\mathbf{r}} \right) = 0. \tag{A23}$$

These equations are related to (23) and (24). Thus, they are a complete system of equations. Finally, if we use Equations (A14)–(A17) and identify $\rho = \mathbf{r}(1 + \varepsilon(\mathbf{r}))$ and $p(\rho) = \mathbf{r}^2\varepsilon'(\mathbf{r})$, the final expressions of the energy-momentum tensors are:

$$T_{\mu\nu} = p(\rho)g_{\mu\nu} + (\rho + p(\rho)) \, U_\mu U_\nu, \tag{A24}$$

$$\tilde{T}_{\mu\nu} = p(\rho)\tilde{g}_{\mu\nu} + \frac{\partial p}{\partial \rho}(\rho)\tilde{\rho}g_{\mu\nu} + \left( \tilde{\rho} + \frac{\partial p}{\partial \rho}(\rho)\tilde{\rho} \right) U_\mu U_\nu$$

$$+ (\rho + p(\rho)) \left( \frac{1}{2} \left( U_\nu U^\alpha \tilde{g}_{\mu\alpha} + U_\mu U^\alpha \tilde{g}_{\nu\alpha} \right) + U_\mu^T U_\nu + U_\mu U_\nu^T \right). \tag{A25}$$

Now, we can use (A24) and (A25) to solve Equations (21)–(24) for a perfect fluid. In this paper, we will use a perfect fluid for the Non-Relativistic and cosmological cases.

**Appendix B. Harmonic Gauge**

We know that the Einstein's equations do not fix all degrees of freedom of $g_{\mu\nu}$. This means that, if $g_{\mu\nu}$ is a solution, then there exists another solution $g'_{\mu\nu}$ given by a general coordinate transformation $x \to x'$. We can eliminate these degrees of freedom by adopting some particular coordinate system, fixing the gauge.

One particularly convenient gauge is given by the harmonic coordinate conditions, which is:

$$\Gamma^\mu \equiv g^{\alpha\beta}\Gamma_{\alpha\beta}^\mu = 0. \tag{A26}$$

Under general coordinate transformation, $\Gamma^\mu$ transform:

$$\Gamma'^\mu = \frac{\partial x'^\mu}{\partial x^\alpha}\Gamma^\alpha - g^{\alpha\beta}\frac{\partial^2 x'^\mu}{\partial x^\alpha \partial x^\beta}.$$

Therefore, if $\Gamma^\alpha$ does not vanish, we can define a new coordinate system $x'^\mu$ where $\Gamma'^\mu = 0$. Thus, it is always possible to choose a harmonic coordinate system (For more details about harmonic gauge, see, for example, [54]).

In the same form, we need to fix the gauge for $\tilde{g}_{\mu\nu}$. It is natural to choose a gauge given by:

$$\tilde{\delta}(\Gamma^\mu) \equiv g^{\alpha\beta}\tilde{\delta}\left(\Gamma_{\alpha\beta}^\mu\right) - \tilde{g}^{\alpha\beta}\Gamma_{\alpha\beta}^\mu = 0, \tag{A27}$$

where $\tilde{\delta}\left(\Gamma^{\mu}_{\alpha\beta}\right) = \frac{1}{2}g^{\mu\lambda}\left(D_{\beta}\tilde{g}_{\lambda\alpha} + D_{\alpha}\tilde{g}_{\beta\lambda} - D_{\lambda}\tilde{g}_{\alpha\beta}\right)$. Then, when we will refer to harmonic gauge, we will use Equations (A26) and (A27).

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
