# Peer review of "δ Gravity: Dark Sector, Post-Newtonian Limit and Schwarzschild Solution"

_universe, doi:10.3390/universe5050096_

Reviewer 1 Report

The revised version justifies with some detail, why delta-gravity could be a good theory, despite of containing ghosts. The authors argue that their proposal is to be considered as a phenomenological theory, and give examples of cases in the literature where this type of instability is handled, in particular in models with phantom energy, as shown end of section 3. Additionally, they make clarifications in several parts of the work. As the main objection was with respect to the instability, I consider that, with these modifications, the work can be published in Universe.

There are several minor redaction problems in the added text (in red). For instance:

The sentence end of section 3, beginning with “Like …”, should be revised.

In the conclusions (added text):

We showed -> We have shown,

phantoms -> phantom

The sentence beginning with “However, some …” is too long and should be split.

Author Response

We are really grateful for your positive comments. According to your remarks, we included in our previous modifications (red) some new ones (in Green).

Reviewer 2 Report

The points raised in my earlier report have been addressed in the revised version. 

Author Response

Thank you very much for your comments.

Reviewer 3 Report

From the formal standpoint all points authors considered are correct, they review published papers and in this sense I have no objections. However the interpretation lacks for the physical justification. I try to explain my point of view.

By the construction authors consider simply the standard first-order perturbation theory for the standard theory GR (general relativity). Then following the modern scientific fashion they pretend that they have deal with new modification of GR. They interpret linear perturbations as new forms of fields (the so-called delta fields). Moreover they assume that delta - fields (say perturbations) remain always in the linear regime and, therefore, they do not interact (any interactions are described by nonlinearity). Though they contribute to the total metric (g + delta-g) and in this sense change trajectories of test particles. They call delta-fields as dark sector.

The fact that delta- fields do not interact (delta-photons, delta-baryons, etc) they can play the role of dark matter. While authors remain in the region where dark matter phenomena have linear character their approach simply coincides with the standard theory in which dark matter is described by additional fields or particles.

However dark matter phenomenon is not linear, e.g., in low surface brightness galaxies the ratio of dark and visible matter components reaches 10^{3}. As far as delta-gravity is concern almost all cosmological scales (save black holes) can be described by the Newton’s gravity, linear metric perturbations. But even in galaxies matter fields (and dark matter) are not in the linear regime.

Authors have some freedom to separate metric and fields on the background fields (they call them standard or ordinary fields) and perturbations (new delta fields). This allows to include even such non-linear phenomena as modern acceleration “effect of dark energy”.

My statement is 1) If authors will insist that delta-fields are only linear, then there are nothing to discuss. This is trivial. All phenomena which show any difference to the standard theory are prescribed to some linear perturbations which are called delta–fields

(such a statement is impossible to verify). Such deviations may be very strong, but for liner theory it is simple and always possible to do (to find such d-fields which obey to linear equations).

2) Authors may consider non-linearity. If they consider higher order perturbations, they simply recover the standard theory. However authors my try to construct the non-linearity in delta- fields only which does not involve (hook) the background fields. In this sense delta-fields will not interact with ordinary background fields. Then such an approach also will not be different from the standard theory GR in which dark matter is simply weakly interacting particles or some other additional fields. Therefore, there will be no new theory as Delta-gravity. All this is just linguistics.

The action suggested by Authors leads to serious problem in quantum field theory, since such an action does not possesses a minimum at all. I do not want to discuss this problem here.

Upon the whole I cannot recommend such a review as a basis of a new physical theory “A Classical Analysis of d Gravity”. It represents a collection of author’s results, but still they have no any relation to some “new theory” at all. Such an interpretation is not justified yet.

Authors my try to change title, accents, etc. then I will have no objections. But in this form this review is misleading.

Author Response

"From the formal standpoint all points authors considered are correct, they review published papers and in this sense I have no objections. However the
interpretation lacks for the physical justification. I try to explain my point of view.

By the construction authors consider simply the standard first-order perturbation theory for the standard theory GR (general relativity). Then following the modern scientific fashion they pretend that they have deal with new modification of GR. They interpret linear perturbations as new forms of fields (the so-called delta fields). Moreover they assume that delta - fields (say perturbations) remain always in the linear regime and, therefore, they do not interact (any interactions are described by nonlinearity). Though they contribute to the total metric (g + delta-g) and in this sense change trajectories of test particles. They call delta-fields as dark sector..."

------------------------------------------------------------------------------------------------------------------------------------------------------------

ANW: We want to quote a comment of a referee to another recently published work on Delta Gravity, that we believe can clarify this point. We are
conscious to be a difficult point:

"...It was somewhat unexpected for me that it is sufficient retaining only the first order perturbation terms in the action to get complete set of
equations of motion for the background and perturbations. The linear term admits both signs and therefore it may look both like a phantom field and the normal field as well.
In general relativity however the negative energy related to perturbations cannot be an arbitrary big. It cannot eat more than all the value related
to the background energy. This gives a very strong restriction which does not allow the acceleration of the present Universe without Lambda term (or dark
energy). It seems if we weaken the above requirement (the above restriction imposed on the value of negative energy), we get indeed some modification of GR."

We believe this is a very good intuition of what is going on. We are not assuming that delta-fields remain always in the linear regime, because then
their size will be bounded and they will not be able to accelerate the expansion of the universe without $\Lambda$. The crucial point is that due to
delta-symmetry, the equations of motion for the background and "perturbations" (delta fields) are a complete set, as the quotation above says.
In a standard perturbation expansion of a non-linear theory, say $y= y_0+y_1+y_2+...$, we get a closed system of equations at each order of perturbation NEGLECTING higher orders. The perturbation expansion is justified only if $y_n
************************************************************************************************************************************************************

"... The fact that delta-fields do not interact (delta-photons, delta-baryons, etc) they can play the role of dark matter. While authors remain in the
region where dark matter phenomena have linear character their approach simply coincides with the standard theory in which dark matter is described by
additional fields or particles.

However dark matter phenomenon is not linear, e.g., in low surface brightness galaxies the ratio of dark and visible matter components reaches 10^{3}.
As far as delta-gravity is concern almost all cosmological scales (save black holes) can be described by the Newton’s gravity, linear metric perturbations.
But even in galaxies matter fields (and dark matter) are not in the linear regime."

------------------------------------------------------------------------------------------------------------------------------------------------------------

ANW: The referee is right here. Certainly in a more elaborated model, people will have to consider self interaction of DM. Our objective is much more
modest in the present paper. In the language of the referee, the regime we consider assume that DM has an equation of state $pressure=0$. We simply noticed that in the Newtonian approximation delta matter density also produces a $1/r$ gravitational potential. So delta matter could be a part, but not all of DM. In fact, with eq. (124) and all Section 4, we want to explain that delta matter amplifies the gravitational effect of ordinary matter, including "Ordinary Dark Matter".
This means that we expect a smaller quantity of Dark Matter, but we need Dark Matter in addition to delta matter. Nevertheless, we must admit that
this is highly speculative at the present stage of the model.

************************************************************************************************************************************************************

Authors have some freedom to separate metric and fields on the background fields (they call them standard or ordinary fields) and perturbations (new
delta fields). This allows to include even such non-linear phenomena as modern acceleration “effect of dark energy”.

My statement is:
1) If authors will insist that delta-fields are only linear, then there are nothing to discuss. This is trivial. All phenomena which show any
difference to the standard theory are prescribed to some linear perturbations which are called delta–fields (such a statement is impossible to verify). Such
deviations may be very strong, but for liner theory it is simple and always possible to do (to find such d-fields which obey to linear equations).

------------------------------------------------------------------------------------------------------------------------------------------------------------

ANW: Please see above the paragraph "We want to quote a comment of a referee..."

************************************************************************************************************************************************************

2) Authors may consider non-linearity. If they consider higher order perturbations, they simply recover the standard theory. However authors my try to
construct the non-linearity in delta-fields only which does not involve (hook) the background fields. In this sense delta-fields will not interact with
ordinary background fields. Then such an approach also will not be different from the standard theory GR in which dark matter is simply weakly
interacting particles or some other additional fields. Therefore, there will be no new theory as Delta-gravity. All this is just linguistics.

------------------------------------------------------------------------------------------------------------------------------------------------------------

ANW: This goes beyond the present paper and it is hard to imagine how the suggestion of the referee will be compatible with delta-symmetry.
We have to remember that the action of delta gravity (equation (16)) is not arbitrary, but is determined by the delta symmetry (equations (14-15)).

************************************************************************************************************************************************************

The action suggested by Authors leads to serious problem in quantum field theory, since such an action does not possesses a minimum at all. I do not
want to discuss this problem here.

Upon the whole I cannot recommend such a review as a basis of a new physical theory “A Classical Analysis of d Gravity”. It represents a collection of
author’s results, but still they have no any relation to some “new theory” at all. Such an interpretation is not justified yet.

Authors my try to change title, accents, etc. then I will have no objections. But in this form this review is misleading."

------------------------------------------------------------------------------------------------------------------------------------------------------------

ANW: Somehow, we feel that this is the crucial reason for the reluctance of the referee to accept the paper. We agree with him that to understand
Delta-Gravity as a "new theory" is not justified yet. We are not claiming that. Our objective is to expose various applications of the model in
cosmological phenomenology and explore its limitations and possible strong points.

For these reasons, we  suggest to modify the title to "Delta Gravity: Dark Sector, Post Newtonian Limit, and the Schwarzschild solution".
Besides, we propose to change the status of the paper from REVIEW to ARTICLE, if this is more acceptable for the referee.

************************************************************************************************************************************************************

Round  2

Reviewer 3 Report

I have no objections against the present manuscript as a regular article. Moreover authors added here new results which can be interpreted in terms of the standard GR. Let us wait for nonlinear effects developed by authors and then we will see, if such an approach suggests indeed some new modification or not.

This manuscript is a resubmission of an earlier submission. The following is a list of the peer review reports and author responses from that submission.

Round  1

Reviewer 1 Report

I have read the manuscript entitled A Classical Analysis of $delta$ Gravity where the authors present a new kind of models and they explored some of their phenomenological consequences. My main concern with this manuscript is the very definition of the theories and how the authors attempt to apply to the gravitational case. If I understand correctly, the theories they consider are described by the action (16). In that Eq. I see some notational inconsistencies (the variations should be applied to functionals like the action and not functions like the Lagrangian), but I understand what the authors mean. Had they used a more proper notation, they would have obtained the integration lacking in Eq (18) and that is what they actually mean by "... it has to be considered as an operation on $\tilde{\Phi}_J$". In any case, the problem I see is when they apply this general definition of the theories to gravity as given in Eqs. (19) and (20). The resulting theory is then a bimetric theory of gravity where the two metric tensors interact non-trivially through a direct kinetic coupling and the matter Lagrangian $\tilde{L}_M$. As it is well-known these theories propagate an Ostrogradski instability (called Boulware-Deser ghost) that make them inconsistent from a theoretical point of view and, also, phenomenologically unappealing (see e.g. Rev.Mod.Phys. 84 (2012) 671-710 or Living Rev.Rel. 17 (2014) 7 for some reviews on the topic). In fact, this has been a very active field of research for the last decade when it was found a particular class of theories that are free from this instability (although cosmological applications were plagued by some other stability and strong coupling problems). An important property of these ghost-free theories is that the interactions are non-derivative, while the interactions in the theory considered by the authors are in fact derivative. However, it has not been found any ghost-free bimetric theory with derivative interactions to date (see  e.g. Class.Quant.Grav. 31 (2014) 165004). The authors have completely neglected all the issues and literature related to massive gravity/bimetric theories. Although all the mathematical developments seem correct, the pathological nature of the starting theory make them lack physical relevance due to the unavoidable presence of a ghost-like degree of freedom and, for this reason, I cannot recommend the present manuscript for publication.

Author Response

REFEREE 1:
"I have read the manuscript entitled A Classical Analysis of $delta$ Gravity where the authors present a new kind of models and they explored some of
their phenomenological consequences. My main concern with this manuscript is the very definition of the theories and how the authors attempt to apply
to the gravitational case. If I understand correctly, the theories they consider are described by the action (16). In that Eq. I see some notational
inconsistencies (the variations should be applied to functionals like the action and not functions like the Lagrangian), but I understand what the
authors mean."...
------------------------------------------------------------------------------------------------------------------------------------------------------------

ANW: You are completely right. Eq. (16), therefore (17) and (18), were modified.

------------------------------------------------------------------------------------------------------------------------------------------------------------
..."Had they used a more proper notation, they would have obtained the integration lacking in Eq (18) and that is what they actually mean by "... it
has to be considered as an operation on $\tilde{\Phi}_J$". In any case, the problem I see is when they apply this general definition of the theories to
gravity as given in Eqs. (19) and (20). The resulting theory is then a bimetric theory of gravity where the two metric tensors interact non-trivially through a
direct kinetic coupling and the matter Lagrangian $\tilde{L}_M$."...
------------------------------------------------------------------------------------------------------------------------------------------------------------

ANW: First of all, delta-gravity is not a bimetric theory in a traditional way as Massive Gravity. Actually, $g$ is used to raise and lower indexes
and it is the only metric term in (20) to define the differential volume. $\tilde{g}$ is introduced as a new tensor field, but it is not a metric. After, it
is demonstrated that $g + \tilde{g}$ is the effective metric, but it is only valided for massless particles. In any case, we have "one metric element". To
emphasize this, we include a paragraph at the end of Section 2 that say:

"At this point, it should be remarked that $\delta$ Gravity is not a bigravity model in a traditional way, because only $g_{\mu \nu}$ is used to raise
and lower indexes, and the differential volume component in (20) just depend on $g_{\mu \nu}$. In fact, it is not a metric model of gravity too because
massive particles do not move on geodesics. Only massless particles move on null geodesics of a linear combination of both tensor fields."

------------------------------------------------------------------------------------------------------------------------------------------------------------

..."As it is well-known these theories propagate an Ostrogradski instability (called Boulware-Deser ghost) that make them inconsistent from a
theoretical point of view and, also, phenomenologically unappealing (see e.g. Rev.Mod.Phys. 84 (2012) 671-710 or Living Rev.Rel. 17 (2014) 7 for some reviews on the topic). In fact, this has been a very active field of research for the last decade when it was found a particular class of theories that are free from this instability (although cosmological applications were plagued by some other stability and strong coupling problems). An important property of these ghost-free theories is that the interactions are non-derivative, while the interactions in the theory considered by the authors are in fact derivative.
However, it has not been found any ghost-free bimetric theory with derivative interactions to date (see e.g. Class.Quant.Grav. 31 (2014) 165004)."...
------------------------------------------------------------------------------------------------------------------------------------------------------------

ANW: Yes, delta-gravity is not a ghost-free model. In fact, we mentioned about the ghost problem in the second paragraph to last in the introduction
and said after (36) that $\tilde{g}$ is the ghost field in our model. Even though ghosts produce a lot of problems, the ghost theories are compatible with
the most classical tests of cosmology based on current data. Probably, this fact was not appropriately exposed. For that reason:

   - We improve the paragraph in the introduction as:
     "We have to say that models like $\delta$ Gravity are not ghost-free, producing some problems like non-unitarity or instabilities [22,38-41].
      Nevertheless, in the current literature, we can see that theories with ghosts are useful if we use them to describe the current data of the most
      classical tests of cosmology [26]-[30]. In the same way, we want to study $\delta$ Gravity as a classical effective model and use it in
Cosmology.
      This means to approach the problem from the phenomenological side instead of neglecting it a priori because it does not satisfy yet all the
      properties of a fundamental quantum theory. Now, the phantom problem is being studied in this moment in $\delta$ Gravity and the results will be
      presented in a future work."

   - To the comment after (36), we modify a sentence by:
     "This result with the fact that the effective metric $\mathbf{g}_{\mu \nu} = g_{\mu \nu} + \tilde{g}_{\mu \nu}$ defines the geometry in our model
say that this particular combination must be seen as the unique graviton [38-41]."

   - We include a paragraph in the conclusions:
     "We showed that $\mathbf{g}_{\mu \nu} = g_{\mu \nu} + \tilde{g}_{\mu \nu}$ is the graviton whereas $\tilde{g}_{\mu \nu}$ is a ghost [19,38-41].
      However, phantoms cosmological models are used to explain the accelerated expansion of the Universe [26]-[30], where the Hamiltonian is not
bounded from below like $\delta$ Theories [22]. However, it is not clear whether this problem will persist or not in a diffeomorphism-invariant model as
      $\delta$ Gravity."

   - For these changes, we include the references [38-41].

------------------------------------------------------------------------------------------------------------------------------------------------------------

..."The authors have completely neglected all the issues and literature related to massive gravity/bimetric theories. Although all the mathematical
developments seem correct, the pathological nature of the starting theory make them lack physical relevance due to the unavoidable presence of a
ghost-like degree of freedom and, for this reason, I cannot recommend the present manuscript for publication."

-----------------------------------------------------------------------------------------------------------------------------------------------------------

ANW: We are conscious of the presence of ghost in the model, presenting the previous changes. However, some solutions to Boulware-Deser ghost problem are been developed (See for instance: arXiv:1711.09009). Particularly in delta-gravity, we have a model with one loop quantum correction only, giving an alternative to avoid the problem. Additionally, the model suggests that $g_{\mu \nu} + \tilde{g}_{\mu \nu}$ is a special combination to define the real graviton. A complete analysis is being developed for a future work. In the meantime, we study delta-gravity as an effective model to apply in
Cosmology, based on the current data, compatible with a phantom component, in a classical level.

Reviewer 2 Report

In this work it is made a revision of a modification of Einstein’s general relativity, called delta-gravity. This modification follows from a proposal for gauge theories whose quantum field theory has only one-loop corrections. The model requires a duplication of the degrees of freedom, and the corresponding terms are introduced as a perturbation. There are new symmetries corresponding to the new degrees of freedom. The Hamiltonian is not bounded from below, fact justified by the authors as a sort of phantom field contributing to dark energy. In the work it is shown, or made plausible, that delta-gravity is consistent with Einstein’s gravity, and with observational results. The metric tensor satisfies Einstein's equations, and the additional degrees of freedom satisfy rather complicated equations. It is mentioned as of particular relevance the fact that this model explains dark energy without cosmological constant. The Newtonian limit seems to be consistent with solar system observations. The interior of black holes has additional structure. Otherwise, as discussed, the duplicated degrees of freedom can enhance the effects of dark matter and inflatons. The authors give analytic solutions and detailed expressions in many cases. The proposal is interesting and could be relevant, as the quantum theory is finite and, so far, its classical predictions are consistent. I recommend its publication after a revision of the redaction.

Author Response

REFEREE 2:
"In this work it is made a revision of a modification of Einstein’s general relativity, called delta-gravity. This modification follows from a proposal
for gauge theories whose quantum field theory has only one-loop corrections. The model requires a duplication of the degrees of freedom, and the
corresponding terms are introduced as a perturbation. There are new symmetries corresponding to the new degrees of freedom. The Hamiltonian is not bounded from below, fact justified by the authors as a sort of phantom field contributing to dark energy. In the work it is shown, or made plausible, that delta-gravity is consistent with Einstein’s gravity, and with observational results. The metric tensor satisfies Einstein's equations, and the additional degrees of freedom satisfy rather complicated equations. It is mentioned as of particular relevance the fact that this model explains dark energy without cosmological constant.
The Newtonian limit seems to be consistent with solar system observations. The interior of black holes has additional structure. Otherwise, as
discussed, the duplicated degrees of freedom can enhance the effects of dark matter and inflatons. The authors give analytic solutions and detailed expressions in many cases.
The proposal is interesting and could be relevant, as the quantum theory is finite and, so far, its classical predictions are consistent. I recommend its
publication after a revision of the redaction."
-----------------------------------------------------------------------------------------------------------------------------------------------------------

ANW: We are really grateful for your positive comments. We made some modifications suggested by the other referees, including a complete revision of the redaction.

Reviewer 3 Report

In their paper the authors introduce the relatively new theory of δ-gravity. They review some of their earlier results on the application of δ-gravity in cosmology and also present some new results, including a study of the classical tests like light deflection and perihelion precession. Overall the paper is well written and gives a detailed review of the subject. The only issue that I found was that the authors have only compared δ-gravity to GR without saying anything about how it fares with other alternative theories of gravity. So I would suggest that they include this in a revised version. Moreover in the paper there are several spelling and grammatical mistakes and so the paper needs substantial text editing. 

Author Response

REFEREE 3:
"In their paper the authors introduce the relatively new theory of d-gravity. They review some of their earlier results on the application of
d-gravity in cosmology and also present some new results, including a study of the classical tests like light deflection and perihelion precession. Overall the
paper is well written and gives a detailed review of the subject. The only issue that I found was that the authors have only compared d-gravity to GR without saying anything about how it fares with other alternative theories of gravity. So I would suggest that they include this in a revised version. Moreover in the paper there are several spelling and grammatical mistakes and so the paper needs substantial text editing."

----------------------------------------------------------------------------------------------------------------------------------------------------------

ANW: Thank you very much for your  comments. Regarding your suggestion, we only compared delta gravity to GR because we are interested to explain the accelerated expansion of the universe without Dark Energy, and many alternative theories of gravity try to emulate this phenomenon. However, we accept that a few comment where we compare delta gravity with other models could be useful to understand and its properties. Considering some mentions in the original version of this paper, we include some comments about bigravity and phantom models, and additionally we mention a method to compare delta-gravity with any other models. Then:

A reader could be thinking that delta-gravity is a bigravity model. However, $\tilde{g}$ is just an additional tensor field without metric properties.
Only $g$ is used to raise and lower indexes and gives the differential volume component. Besides, we have only one component to define the graviton. To
avoid this:

   - At the end of Section 2, we include a paragraph that say:
     "It should be remarked that $\delta$ Gravity is not a bigravity model in a traditional way, because only $g_{\mu \nu}$ is used to raise and lower
      indexes, and the differential volume component in (20) just depend on $g_{\mu \nu}$. In fact, it is not metric model of gravity too because
massive particles do not move on geodesics. Only massless particles move on null geodesics of a linear combination of both tensor fields."

   - Besides, we include new references ([38-41]) and a comment after (36) to modify a sentence by:
     "This result with the fact that the effective metric $\mathbf{g}_{\mu \nu} = g_{\mu \nu} + \tilde{g}_{\mu \nu}$ defines the geometry in our model
      say that this particular combination must be seen as the unique graviton [38-41]."

The phantom model are highly compatible with the most classical tests of cosmology according to the current data, so they are excellent effective
models. This means that a model is phantom-like, then it could be compatible with the data. delta-gravity is the case. To explain this:

   - We have in the introduction the paragraph:
     "We have to say that models like $\delta$ Gravity are not ghost-free, producing some problems like non-unitarity or instabilities [22,38-41].
      Nevertheless, in the current literature, we can see that theories with ghosts are useful if we use them to describe the current data of the most
      classical tests of cosmology [26]-[30]. In the same way, we want to study $\delta$ Gravity as a classical effective model and use it in Cosmology.
      This means to approach the problem from the phenomenological side instead of neglecting it a priori because it does not satisfy yet all the
      properties of a fundamental quantum theory. Now, the phantom problem in $\delta$ Gravity is being studied in this moment and the results will be
      presented in a future work."

  - In section 3.4, we introduce $\delta$ inflation. In a particular case, a effective Hubble parameter is defined in eq. (80). From this expression,
we can define an effective "equation of state parameter" $\omega_{eff}$. If $\omega_{eff} < -1$, the expansion is phantom-like. To explain this, we
    include a new reference ([48]), where $\delta$ Inflation is introduce with a little more details. Besides, eq. (81) is included and the next paragraph is introduced at the end of section 3.4:
    "If we apply (80) in (81), we can study the expansion behavior of our model. If $\omega_{eff}(t) < -1$, the expansion is like a phantom model. This
     calculation is briefly developed in [48], where we demonstrated that $\delta$ Gravity works like a phantom model. Like an additional point, any
     alternative theories of gravity can be compared with $\delta$ Gravity in a cosmological level if an $\omega_{eff}(t)$ can be defined. A more
detailed version of this work is in progress."

Finally, we made a complete revision of the redaction to correct  spelling and grammatical mistakes.      

Round  2

Reviewer 1 Report

I  have read the authors response to my first report and, in view of the provided answers to my points, I must stand by my recommendation against publication. Some comments I would like to make are:

1. The integration is still missing in (18).

2. The proposed theory IS a bi-metric theory. Perhaps the authors have a non-standard understanding of what a bi-metric theory is, but, as they authors point out, there are two rank-2 tensors. Whether one only uses one of them to raise or lower indices, or particles follow or not the corresponding geodesics, etc. is irrelevant. The important feature is that there are two propagating spin-2 particles that are coupled to each other.

3. It is not only that ghosts cause problems, it is that ghosts make theories phenomenologically inconsistent. The authors seem to confuse the presence of ghosts with a phantom behaviour. Though related, these two things are not the same. In fact, it is possible to have phantom behaviour without ghosts. This is what has been considered in the literature and, in particular, in the Refs provided by the authors. It is remarkable that Ref. [29] precisely puts a very stringent constraint on the cut-off scale of a possible ghost. Ghosts theories are not compatible with the most classical tests of cosmology based on current data, among other things, because the very presence of a ghost-like instability makes the background cosmology unstable and, therefore, one cannot trust such background solutions.